# Autocrine VEGF drives neural stem cell proximity to the adult hippocampus vascular niche

Tyler J Dause[1], Jiyeon K Denninger[1], Robert Osap[1], Ashley E Walters[1], Joshua D Rieskamp[2], Elizabeth D Kirby[1,3]

**The vasculature is a key component of adult brain neural stem cell (NSC) niches. In the adult mammalian hippocampus, NSCs reside in close contact with a dense capillary network. How this niche is maintained is unclear. We recently found that adult hippocampal NSCs express VEGF, a soluble factor with chemoattractive properties for vascular endothelia. Here, we show that global and NSC-specific VEGF loss led to dissociation of NSCs and their intermediate progenitor daughter cells from local vasculature. Surprisingly, though, we found no changes in local vascular density. Instead, we found that NSC-derived VEGF supports maintenance of gene expression programs in NSCs and their progeny related to cell migration and adhesion. In vitro assays revealed that blockade of VEGF receptor 2 impaired NSC motility and adhesion. Our findings suggest that NSCs maintain their own proximity to vasculature via self-stimulated VEGF signaling that supports their motility towards and/or adhesion to local blood vessels.**

## Introduction

Neural stem cells (NSCs) and their progeny reside in two primary neurogenic niches in the adult mammalian brain, the subventricular zone (SVZ) and the dentate gyrus (DG) of the hippocampus. Within the adult DG niche, NSCs give rise to intermediate progenitor cells (IPCs) that go on to generate new neurons, a process that is conserved across most land-based mammals studied to date (Kempermann, 2015; Charvet & Finlay, 2018). Evidence in rodent models supports the functionality of adult neurogenesis, showing that newborn neurons can modulate DG circuit properties (McAvoy & Sahay, 2017; Tuncdemir et al, 2019; Li et al, 2023) and support spatial learning and memory and affective behaviors (Miller & Sahay, 2019). Adult neurogenesis has also generated interest as a potential therapeutic target in a number of diseases and disorders in which hippocampal memory function is impaired (McAvoy & Sahay, 2017; Wander & Song, 2021; Choi & Tanzi, 2023).

The vasculature is a key component of stem cell niches throughout the body (Gómez-Gaviro et al, 2012; He et al, 2014; Verma et al, 2018), including the adult brain niches (Palmer et al, 2000; Shen et al, 2008; Tavazoie et al, 2008; Sun et al, 2015). The adult DG vascular niche is characterized by a dense network of planar capillaries concentrated in the subgranular zone (SGZ) where NSCs and their daughter IPCs (together, NSPCs) reside. Within the SGZ, IPC bodies make close contact with local vessels, using the vessels to guide their migration tangentially through the SGZ away from their parent NSCs before differentiating into a neuronal phenotype (Palmer et al, 2000; Sun et al, 2015). NSCs also achieve unique vascular contact via their radial glia-like process, which extends through the granule cell layer to wrap around vessels in the molecular layer (Moss et al, 2016; Licht et al, 2020). This vascular niche is hypothesized to provide NSPCs with access to circulating factors, as well as signaling from the endothelial cells that comprise vessels, both of which support NSC maintenance and neurogenesis (Villeda et al, 2014; Licht & Keshet, 2015; Yousef et al, 2019; Kim et al, 2021). Despite wide-spread recognition of the importance of the adult DG vascular niche, there is little known about what signals maintain it.

We previously showed that adult DG NSCs synthesize vascular endothelial growth factor (VEGF) (Kirby et al, 2015), a pleiotropic soluble protein with multiple signaling roles in the adult brain (Lange et al, 2016). We showed that self-produced NSPC-VEGF is necessary for maintenance of the NSPC pool (Kirby et al, 2015) and suggested that this may rely on autocrine signaling via VEGF receptors expressed on NSCs. However, VEGF also can have potent mitogenic and chemoattractant roles for vascular endothelial cells. The role of NSPC-VEGF in DG vascular niche maintenance was left largely unexplored in our previous work.

In this study, we investigated the hypothesis that NSPCs maintain their own vascular niche in the adult mouse DG via

[1]Department of Psychology, College of Arts and Sciences, The Ohio State University, Columbus, OH, USA  [2]Neuroscience Graduate Program, The Ohio State University, Columbus, OH, USA  [3]Chronic Brain Injury Program, The Ohio State University, Columbus, OH, USA

Correspondence: kirby.224@osu.edu
Tyler J Dause's present address is Novasenta, Pittsburgh, PA, USA
Jiyeon K Denninger's present address is Sarepta Therapeutics, Cambridge, MA, USA
Robert Osap's present address is Ohio University Medical School (student), Athens, OH, USA
Joshua D Rieskamp's present address is RWC Global, Atlanta, GA, USA

production of VEGF. In support of this hypothesis, we found that loss of NSPC-specific VEGF led to disruption of the neurovascular niche. Unexpectedly, however, niche disruption was not accompanied by detectable changes in vessel density, vascular endothelial survival, angiogenesis, or changes to blood–brain barrier (BBB) proteins. Instead, VEGF loss caused impaired expression of genes related to motility/adhesion in NSCs in vivo and impaired both motility and adhesion of NSCs in vitro. Overall, our findings suggest that NSPC-derived VEGF supports NSPC proximity to vessels via self-signaling.

# Results

### Hippocampal NSPCs and astrocytes express VEGF

Our first step to better understand how NSPC-derived VEGF might affect the neurogenic vascular niche was to more fully characterize VEGF production by different DG cell types in vivo. We previously showed qualitative presence of a VEGF transcriptional reporter (VEGF-GFP) in DG radial glia-like NSCs (RGL-NSCs) and IPCs, as well as in other DG cells (Kirby et al, 2015). Consistent with VEGF being expressed by NSPCs as well as other DG cell types, we also showed that knockout of VEGF in NSPCs in adulthood led to a significant but partial decrease in whole DG VEGF levels at the RNA and protein levels. To expand on these previous findings here, we first quantified all the VEGF-expressing populations of the DG using a VEGF-GFP transcriptional reporter mouse (Fig 1A). We found VEGF-GFP expression in the granule cell layer and SGZ primarily in GFAP$^+$/SOX2$^+$ RGL-NSCs and GFAP$^-$/SOX2$^+$ IPCs, whereas VEGF-GFP expression in the hilus was mostly in GFAP$^+$/SOX2$^+$ stellate astrocytes (Fig 1B and C). Nearly all astrocytes, RGL-NSCs, and IPCs were VEGF-GFP$^+$, with little expression in doublecortin$^+$ (DCX$^+$) neuronally committed neuroblasts and immature neurons (INs) (Fig 1D). We also confirmed VEGF-GFP expression in NSCs using co-labeling for Nestin, a well-established marker of RGL-NSCs (Fig 1E).

To explore relative VEGF expression in a more unbiased manner, we used published single-cell and bulk RNA sequencing datasets derived from the adult mouse DG. We found that RGL-NSCs and their IPC progeny expressed Vegfa at levels intermediate between two other known producers of VEGF, astrocytes and endothelia (Fig S1A) (Hochgerner et al, 2018; Batiuk et al, 2020; Walker et al, 2020). Analysis of additional published RNAseq datasets revealed that hippocampal NSC Vegfa expression increases 1,285 ± 45% from embryonic to adult ages (Berg et al, 2019) (Fig S1B) and is 1,254 ± 102% greater in hippocampal NSPCs than in NSPCs from the SVZ (Adusumilli et al, 2021) (Fig S1C). To further confirm these sequencing-based results, we performed RNAscope in situ hybridization in adult mouse fixed tissue sections. We used GFAP immunolabeling to identify putative RGL-NSCs and astrocytes based on their unique morphologies. GFAP$^+$ putative RGL-NSCs expressed Vegfa but 62.43 ± 4.62% less than putative GFAP$^+$ astrocytes (Fig 1F and G). Taken together, these data further validate our previous findings that NSPCs are a significant source of VEGF in the adult mouse DG, although they produce less Vegfa per cell at the RNA-level than astrocytes.

### Broad loss of VEGF disrupts the DG neurogenic vascular niche

Prenatally, brain-derived VEGF serves as a chemoattractant and mitogen for endothelial cells of developing vasculature, drawing in nascent blood vessels to vascularize the central nervous system (Gerhardt et al, 2003; Haigh et al, 2003; Raab et al, 2004). Adult brain vasculature, however, is frequently described as quiescent and VEGF-refractory, obtaining an independence from the reliance on VEGF signaling that characterizes developing vasculature (Wälchli et al, 2023). To broadly investigate whether VEGF had any role in regulating the adult DG vascular niche, we used lentiviral expression of an shRNA against Vegfa (or a scramble control). We perfused mice 21 d after stereotaxic viral infusion in the adult DG (Fig 2A). The lentiviral vectors showed wide tropism, as expected (Fig 2B), and Vegfa shRNA reduced VEGF immunoreactivity throughout the DG by 74 ± 12% compared with the scramble control (Scramble: 3.820 ± 0.3622, Vegfa: 0.4395 ± 0.2778, mean ± SEM). We next quantified the shortest distance between CD31$^+$ vasculature and the cell body center of GFP$^+$/SOX2$^+$/GFAP$^+$ RGL-NSCs and GFP$^+$/SOX2$^+$/GFAP$^-$ IPCs (Fig 2C–E). These distances were compared with the distance of SGZ cells sampled randomly (points every ~100 $\mu$m along the SGZ midline, throughout the SGZ) to reveal whether cells show preferential association with vessels (closer than random) or dissociation (farther than random). As expected, IPCs were generally closer than random to CD31$^+$ vessels, whereas RGL-NSC somas were farther than random. Vegfa shRNA expression led to a significant disruption of vessel association in both GFP$^+$SOX2$^+$GFAP$^+$ RGL-NSCs and GFP$^+$SOX2$^+$GFAP$^-$ IPCs. RGL-NSCs in iKD mice were 3.112 ± 0.6886 $\mu$m farther from the nearest vessel than those in WT mice. IPCs in iKD mice were 3.565 ± 0.8309 $\mu$m farther from the nearest vessel than IPCs in WT mice, residing a similar distance as random cells. No changes in CD31$^+$ vessel density were found in any of the DG layers (Fig 2F and G). These findings suggested that RGL-NSC and IPC proximity to local vessels relies on VEGF expression in the DG but that this reliance is not driven by changes in vessel survival. Given the wide tropism of the lentiviral tool used, however, whether NSPC-derived VEGF was sufficient to drive this effect remained unclear.

### NSPC-VEGF loss disrupts the DG neurogenic vascular niche

To examine the role of NSPC-derived VEGF in maintaining NSPC-vessel proximity, we next used a NSPC-specific, inducible VEGF knockdown model. To knockdown VEGF specifically in NSCs and their progeny, we used NestinCreER$^{T2+/-}$; Vegfa$^{lox/lox}$; ROSA-LoxSTOPLox-EYFP$^{+/+}$ mice. We treated NestinCreER$^{T2+/-}$; Vegfa$^{lox/lox}$; ROSA-LoxSTOPLox-EYFP$^{+/+}$ (iKD) mice and NestinCreER$^{T2+/-}$; Vegfa$^{wt/wt}$; ROSA-LoxSTOPLox-EYFP$^{+/+}$ (WT) littermates with tamoxifen (TAM) to induce recombination and EYFP expression in NSCs, IPCs and their progeny (Fig 3A). Our own work (Kirby et al, 2015; Dause & Kirby, 2020; Smith et al, 2022) and work by others (Lagace et al, 2007; Sun et al, 2014) have shown that this NestinCreER$^{T2}$ line drives loxP recombination with high specificity in NSCs and IPCs. Although we were not able to verify VEGF knockdown in individual NSPCs here due to technical limitations, we have previously shown that TAM administration in these mice results in loss of about 1/3 of total DG VEGF compared with WT littermates (Kirby et al, 2015).

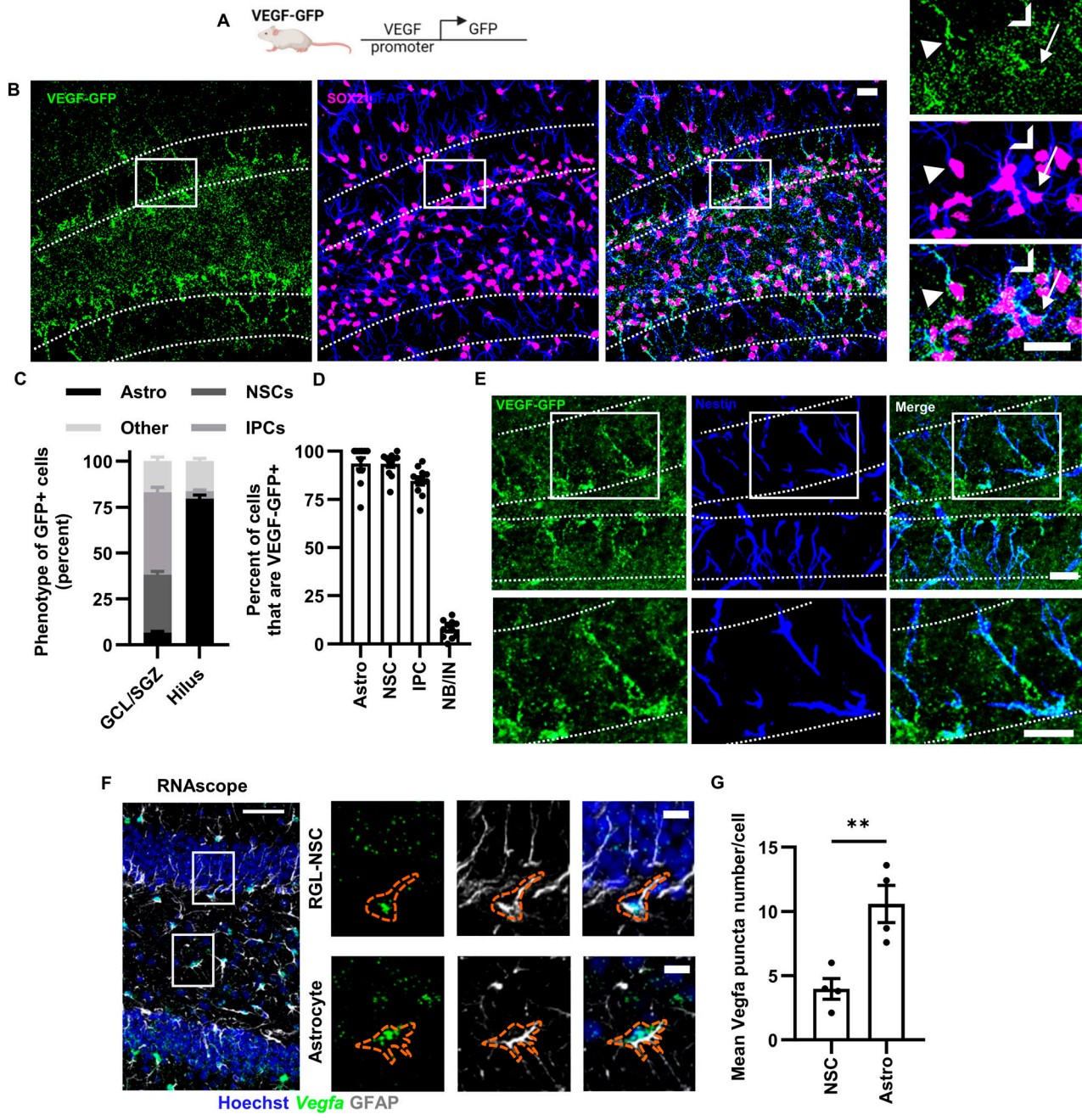

**Figure 1. DG NSPCs and astrocytes express VEGF.**
**(A)** Diagram of VEGF-GFP mouse model. **(B)** Representative immunofluorescent images of GFP, GFAP, and SOX2 in the DG. Dashed line shows granule cell layer. Solid box shows origin of subset images to the right of GFP⁺/GFAP⁺/SOX2⁺ RGL-NSCs (arrowhead), GFP⁺/GFAP⁻/SOX2⁺ IPCs (chevron) and GFP⁺/GFAP⁺/SOX2⁺ astrocytes (arrow). **(C)** Percent of GFP⁺ cells of each phenotype: Astrocytes, RGL-NSCs, IPCs and other. N = 11 mice. **(D)** Percent of Astrocytes, NSCs, IPCs, and NB/IN that were GFP⁺. N = 11 mice. **(E)** Representative immunofluorescent image of GFP and Nestin immunolabeling in the DG of VEGF-GFP mice. Lower images show detail within box outlined in top images. Dashed line shows granule cell layer. **(F)** Vegfa in situ hybridization co-labeled with GFAP antibody in adult mouse DG. Subset images of Vegfa in situ hybridization co-labeled with GFAP antibody in adult mouse DG on the right. **(G)** Mean Vegfa RNA per cell in RGL-NSCs and astrocytes. N = 4 mice. Mean ± SEM plus individual mice shown throughout. Scale bars represent (B, E) 20 μm, (F) 50 μm, subset 10 μm *P < 0.05; **P < 0.01.

We next quantified the shortest distance between CD31⁺ vasculature and the cell body center of EYFP⁺/GFAP⁺ RGL-NSCs and EYFP⁺/MCM2⁺ IPCs (Fig 3B). Among WT mice, we found, as expected, that RGL-NSC somas were farther than random from CD31⁺ vessels, whereas IPCs were closer to vessels than random distribution (Fig 3C–F). In iKD mice, both RGL-NSC and IPC somas were significantly farther from the vasculature compared with WT controls (Fig 3C–F). RGL-NSC somas in iKD mice were 2.568 ± 1.197 μm farther from vessels than those in WT mice. IPCs in iKD mice were 1.579 ± 0.6776 μm farther from the nearest vessel than IPCs in WT mice,

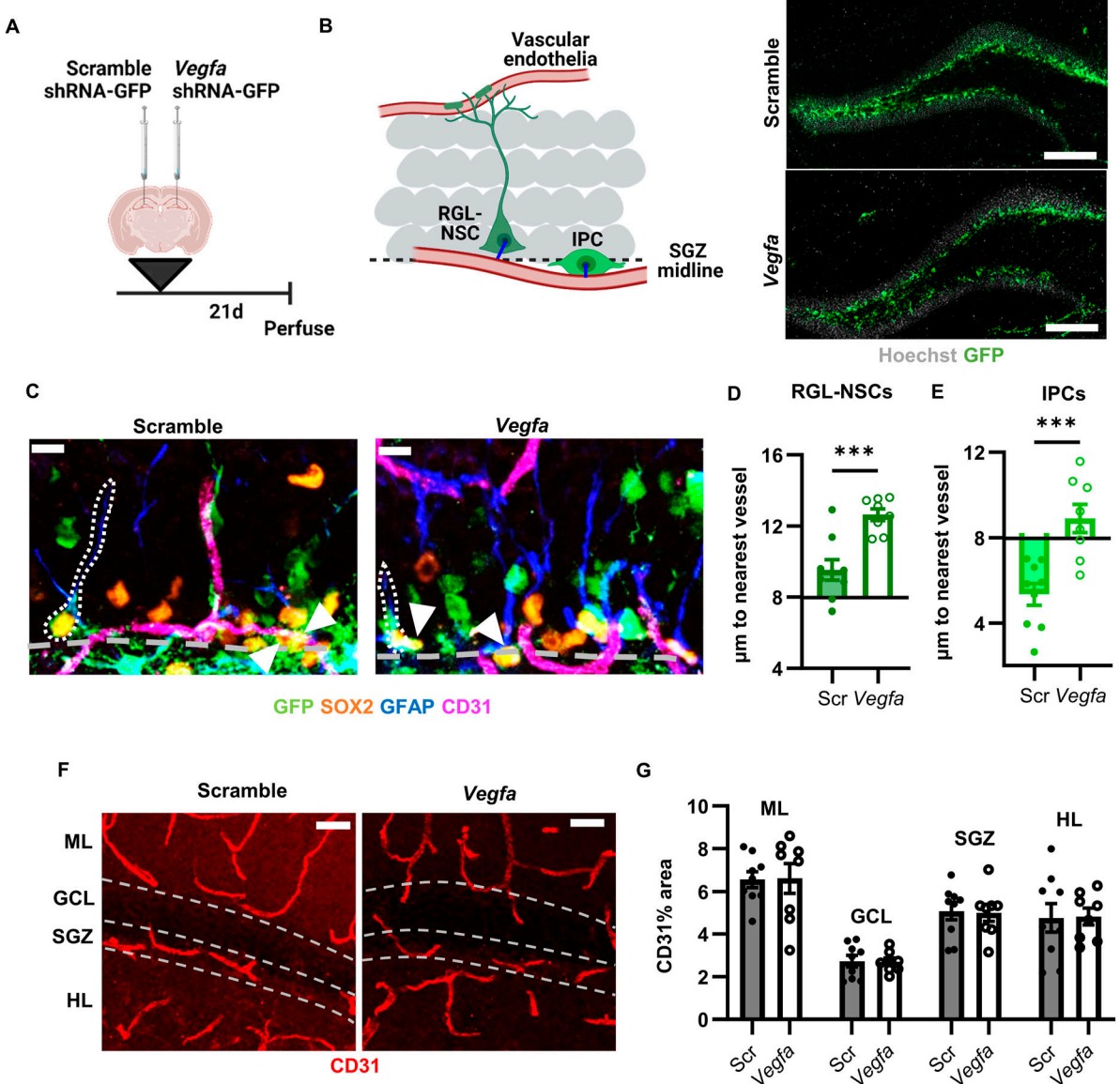

**Figure 2. NSPC proximity to vessels is disrupted by broad shRNA-mediated VEGF knockdown.**
**(A)** Diagram of experimental design. **(B)** Representative images of Scramble or Vegfa shRNA infection (GFP⁺) in the DG (Hoechst shows cell nuclei) 21 d after viral infusion. **(C)** Representative immunofluorescent images of shRNA expressing (GFP⁺) GFAP⁺SOX2⁺ RGL-NSCs and GFAP-SOX2⁺ IPCs and their association with the CD31⁺ vasculature 21 d after viral infusion. White dashes outline GFP⁺ RGL-NSCs, arrowheads indicate GFP⁺ IPCs. SGZ midline shown as grey dashed line. **(D, E)** Distance from nearest CD31⁺ vessel for WT or iKD GFP⁺ or GFP⁻ GFAP⁺SOX2⁺ RGL-NSCs (D) or GFAP⁻SOX2⁺ IPCs (E) 21 d after viral infusion. Bars start at average distance for a random SGZ cell. N = 8 mice/group. Mean ± SEM plus individual mice shown. **(F)** Representative immunofluorescent images of CD31⁺ endothelia in the DG subregions. Dashed lines indicate borders between subregions. **(G)** Comparison of CD31 percent area in the DG subregions in scramble shRNA and Vegfa shRNA infused mice 21 d after surgery. Scale bars represent (B) 200 μm, (C) 10 μm, (F) 50 μm. ML, molecular layer; GCL, granule cell layer; SGZ, subgranular zone; HL, hilus. ***$P < 0.001$.

again leaving these iKD cells at a similar distance as random cells, similar to our findings with more general VEGF knockdown via shRNA. Also similar to IPCs as a whole, proliferative cells labeled with the mitotic marker BrdU 2 h before perfusion were closer to vessels than random in WT mice and this association was disrupted in VEGF iKD mice (Fig 3G and H). We also found that RGL-NSC process contact with vessels was significantly disrupted 21 d after TAM (Fig 3I and J). These data show significant disruption of vascular proximity of NSPCs after loss of NSPC-derived VEGF.

We next examined neuroblast and immature neuron progeny of NSPCs. These cell types do not normally self-synthesize notable quantities of VEGF (Figs 1D and S1A). We categorized DCX⁺ cells into neuroblasts and immature neurons morphologically, similarly to previously published methods (Plümpe et al, 2006). Similar to previous work (Sun et al, 2015), we found that immature neurons were generally farther from vessels than neuroblasts (Fig 3K–M). There was no difference in vessel association of neuroblasts or immature neurons between WT and iKD mice. All together, these data suggest that NSPC-specific VEGF is necessary to maintain the

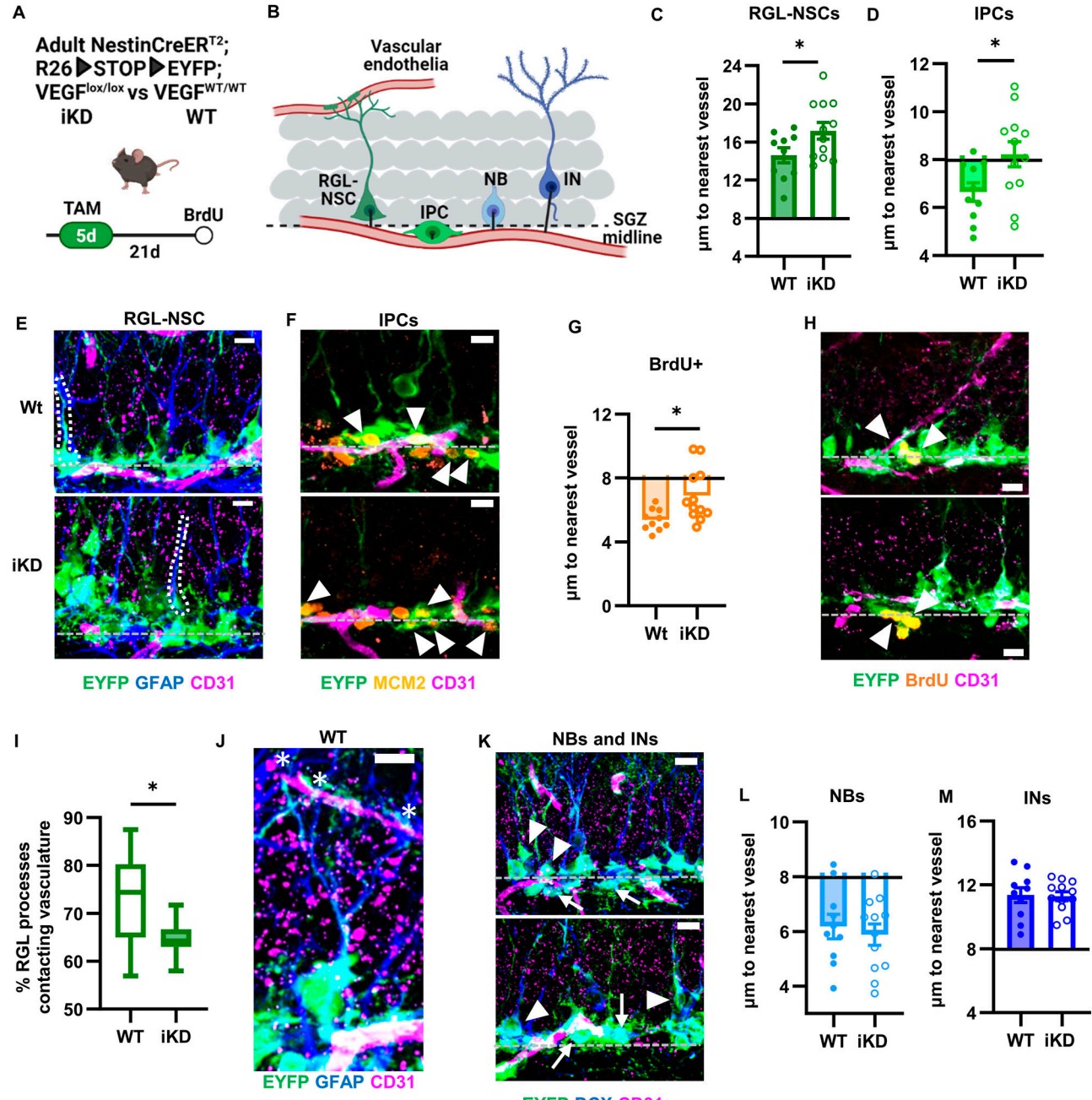

**Figure 3. NSPC proximity to vessels is disrupted by induced NSPC-VEGF loss.**
**(A)** Diagram of experimental design and timeline. **(B)** Diagram of vascular distance measurements in NSCs and progeny. **(C)** Distance from nearest CD31+ vessel for WT or iKD RGL-NSCs in SGZ. Bars start at average distance for a random SGZ cell. **(D)** Distance from nearest CD31+ vessel for WT or iKD IPCs in SGZ. Bars start at average distance for a random SGZ cell. **(E)** Representative immunofluorescent images of GFAP+EYFP+ RGL-NSCs (white dashed outline) and CD31+ endothelia 21 d after NSPC-VEGF knockdown. **(F)** Representative immunofluorescent images of MCM2+EYFP+ IPCs (arrowheads) and CD31+ endothelia 21 d after NSPC-VEGF knockdown. **(G)** Distance from nearest CD31+ vessel for WT or iKD BrdU+EYFP+ cells in SGZ. Bars start at average distance for a random SGZ cell. **(H)** Representative immunofluorescent images of BrdU+EYFP+ cells and CD31+ endothelia 21 d after NSPC-VEGF knockdown. Chevrons indicate BrdU+EYFP+ cells. **(I)** Comparison of the percent of RGL-NSCs with a radial process contacting the vasculature. **(J)** Representative immunofluorescent image of GFAP+EYFP+ RGL-NSC radial process contacting CD31+ endothelia in a WT mouse. White * indicate points of putative contact. Arrow indicates vascular contact. **(K)** Representative immunofluorescent images of DCX+EYFP+ neuroblasts (NB, horizontal morphology, arrows), immature neurons (IN, dendritic morphology, arrowheads) and CD31+ endothelia 21 d after NSPC-VEGF knockdown. **(L)** Distance from nearest CD31+ vessel for WT or iKD DCX+ NBs in SGZ. Bars start at average distance for a random SGZ cell. **(M)** Distance from nearest CD31+ vessel for WT or iKD DCX+ INs in SGZ. Bars start at average distance for a random SGZ cell. Mean ± SEM plus individual mice shown throughout. N = 10 WT, 12 iKD. Scale bars all represent 10 $\mu$m. *$P$ < 0.05. (E, F, H, K) Grey dashed line indicates SGZ midline.

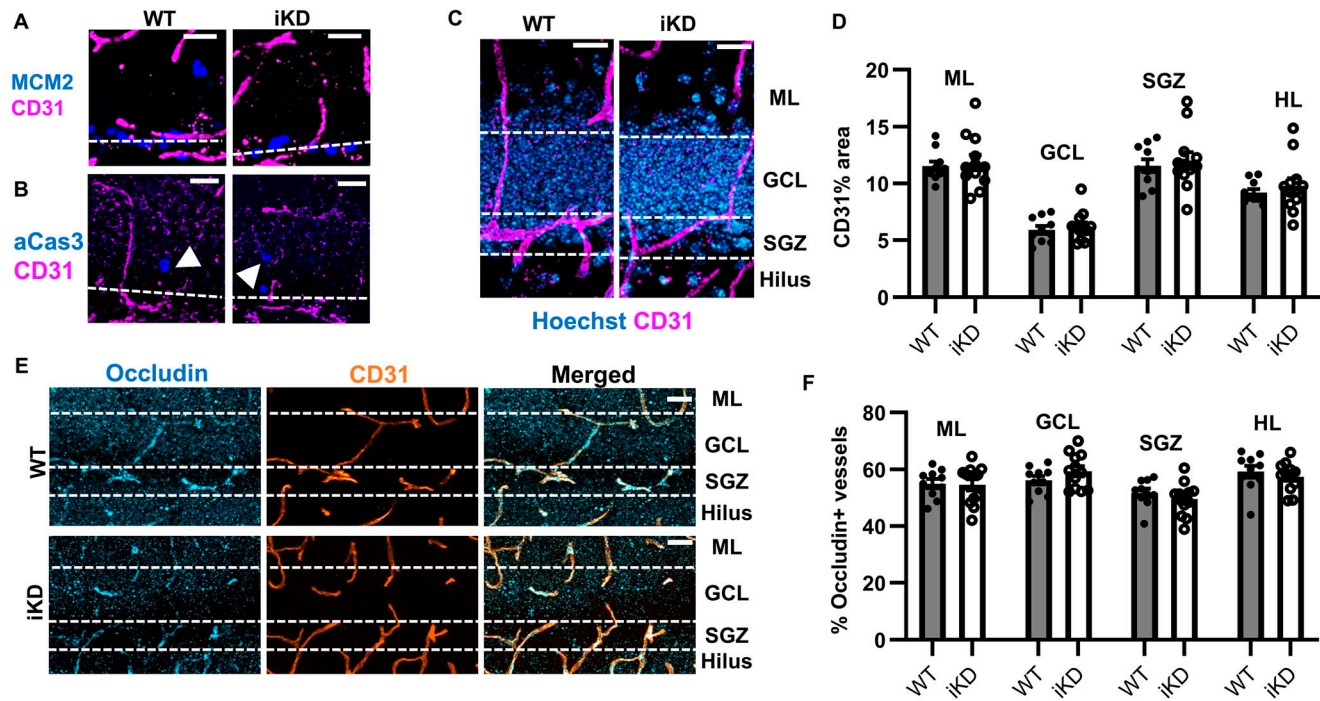

**Figure 4. NSPC-VEGF knockdown does not detectably alter the DG vasculature.**
**(A)** Representative immunofluorescent images of MCM2+ cells and CD31+ endothelia in WT and iKD mice. Dashed lines represent granular cell layer. **(B)** Representative immunofluorescent images of aCas3+ cells and CD31+ endothelia in WT and iKD mice. Arrowheads indicate aCas3+ cell. Dashed lines represent granular cell layer. **(C)** Representative immunofluorescent images of CD31+ endothelia in the DG subregions after NSPC-VEGF knockdown. Hoechst used to label cell bodies. Dashed lines indicate borders between subregions. **(D)** Comparison of CD31 percent area in the DG subregions in WT and iKD mice 21 d after TAM. **(E)** Representative immunofluorescent images of occludin+CD31+ endothelia in the DG subregions 21 d after TAM. Dashed lines indicate borders between subregions. **(F)** Comparison of percent of CD31+ area occupied by occludin labeling in the DG subregions in WT and iKD mice 21 d after TAM. Mean ± SEM shown throughout. N = 10 WT, 12 iKD mice. Scale bars represent 20 μm.

vascular proximity of NSPCs, with no effect on their neuroblast and immature neuron progeny.

## NSPC-VEGF loss has no detectable effect on DG vasculature

We next investigated whether loss of NSPC-VEGF may disrupt maintenance of vasculature in the DG cell layers, specifically hypothesizing that local changes in endothelial proliferation, survival, or migration may underlie the loss of NSPC-vessel proximity we observed. First, we examined the effect of NSPC-VEGF loss on angiogenesis by co-labeling CD31+ endothelia with MCM2, a marker of the cell cycle (Krstulja et al, 2008). In adulthood, brain vasculature is generally considered quiescent (nonproliferative and stable) (Wälchli et al, 2023). Consistent with this nonproliferative nature of adult vasculature, we found no evidence of angiogenesis in the adult mouse DG in both WT and iKD mice (Fig 4A). We also found no evidence of endothelial cell death, as evidenced by a lack of co-labeling for CD31 with activated caspase-3, a marker of apoptosis (Fig 4B). Finally, we quantified vascular coverage to determine if NSPC-VEGF loss drove endothelial regression away from the SGZ. Using CD31+ immunolabeled thresholded area analysis, we found that there was no difference between WT and iKD mice in CD31+ area in any DG subregion (Fig 4C and D). These findings suggested no broad vascular gain, loss or migration after NSPC-VEGF loss.

In addition to its role in regulating endothelial migration and proliferation, VEGF also stimulates shedding of BBB associated proteins on endothelial cells, thereby increasing BBB permeability (Zhang et al, 2000). Occludin is a tight junction membrane protein and a structural component of the BBB that plays an integral role in maintaining BBB impermeability. We therefore used co-labeling of occludin on CD31+ vessels as a measure of BBB integrity (Fig 4E). We found significant, though small, regional differences in the percent of occludin co-labeling on CD31+ vessels, with SGZ being the region with the least co-labeling (see Table S2). However, there was no difference in the percentage of occludin+ vessels between WT and iKD mice in any DG subregion (Fig 4F). Taken together, our data were contrary to our expectations and revealed no measurable impact of NSPC-VEGF loss on multiple aspects of adult DG vasculature.

## Loss of NSPC-VEGF disrupts gene expression programs regulating cell adhesion/migration

Given the apparent lack of change in adult DG vasculature after NSPC-VEGF loss, we next examined changes in NSPCs themselves in more detail. We performed single-cell RNA sequencing of RGL-NSCs and their progeny acutely isolated from adult mouse DG of our inducible, NSPC-specific VEGF knockdown model (Fig 5A). 21 d after TAM, EYFP+ cells were acutely isolated from dissected DGs using fluorescence activated cell sorting and RNA from single cells was sequenced using a 10X Chromium platform. In total, 5,543 WT EYFP+

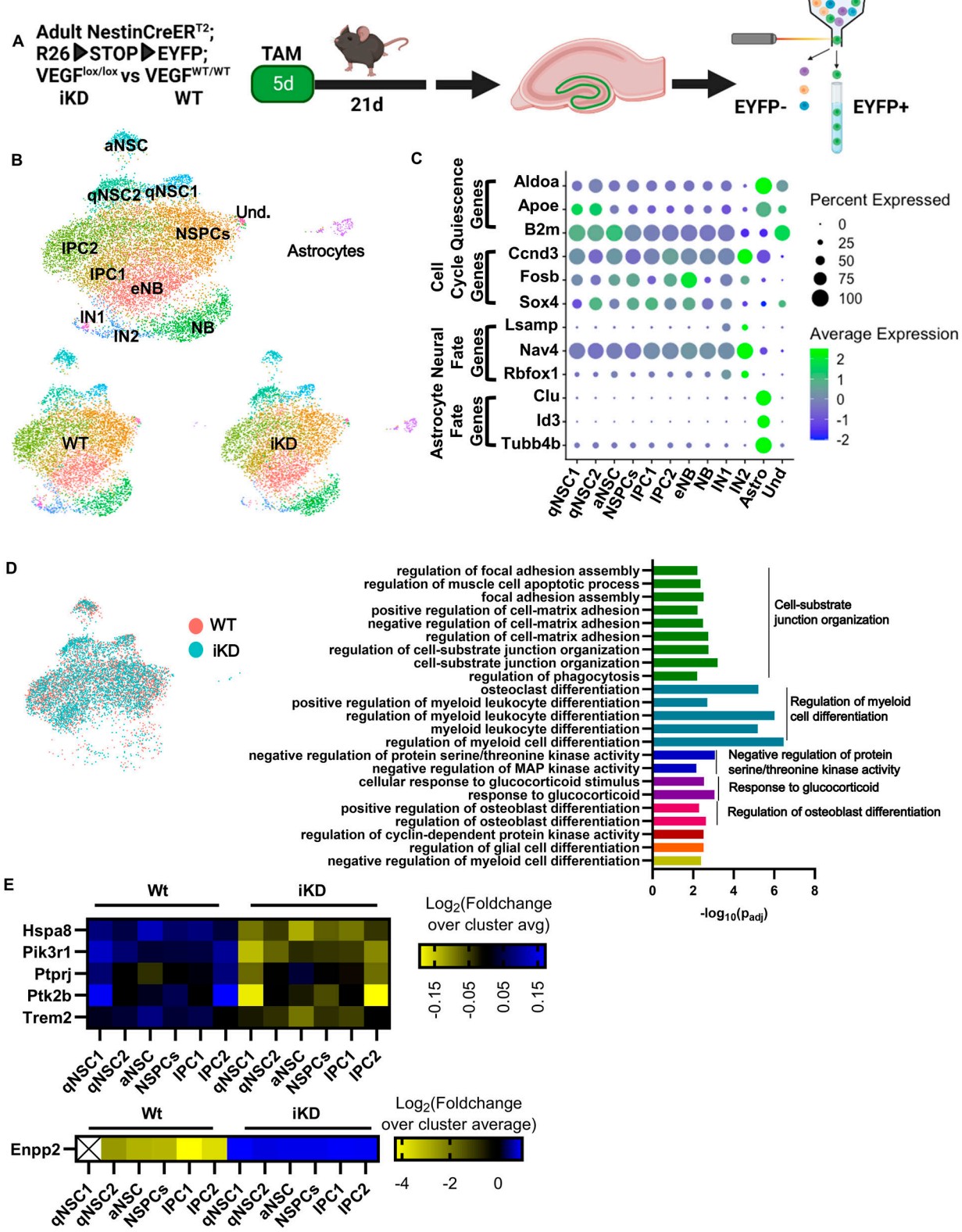

**Figure 5. NSPC-VEGF regulates genes related to cell adhesion in vivo.**
**(A)** Diagram of experimental design and timeline. N = 5 mice/genotype were pooled and sorted. **(B)** UMAP of WT and iKD cells yielded 11 subpopulations. Clusters are represented by different colors and phenotypes were assigned by gene expression and GO analysis of DEGs between clusters. **(C)** Dot plot visualization of average expression and percent of cells expressing genes related to quiescence, the cell cycle, neurogenic, and gliogenic fate. Average expression is a z-score. **(D)** UMAP showing

cells and 5,261 iKD EYFP⁺ cells met inclusion criteria after being captured and sequenced. Uniform manifold approximation and projection (UMAP) for dimension reduction analysis revealed that both WT and iKD cells were present in 11 different subpopulations characterized by gene expression profiles linked to Gene Ontology (GO) terms consistent with cell types across the neurogenic cascade (Fig 5B and Table S1).

To confirm cluster identities, we examined both GO terms associated with cluster marker genes and the expression of known markers of quiescence, the cell cycle, neural fate, and gliogenic fate. We identified three clusters of NSCs (10.17% of cells), five clusters containing IPCs or NBs (85.09% of cells), two neuronal clusters (3.13% of cells), and one astrocytic cluster (1.24% of cells). The NSC clusters were generally characterized by high expression of quiescence-related genes such as Aldoa, Apoe, and B2m and GO terms associated with ribosomal biogenesis, a process previously shown to be high in NSCs (Shin et al, 2015; Harris et al, 2021) (Fig 5C). One of the NSC clusters was characterized by GO terms associated with G1/S transition and cell cycle activation, leading us to define these as active NSCs (Fig 5C). The IPC and NB clusters showed progressive loss of quiescence gene expression coupled with expression of cell cycle–related genes such as Ccnd3, Fosb, and Sox4 (Fig 5C). GO terms in these populations also highlighted cell cycle–related processes such as regulation of G1/S transition of mitotic cell cycle. We defined one population as a putative mix of NSCs and IPCs (NSPCs) because it showed gene expression and GO terms intermediate between the active NSC and IPC clusters (Fig 5C). The two IN populations showed strong expression of neuronal fate genes and GO terms associated with neuronal differentiation. The astrocyte cluster had high expression of quiescence genes coupled with high expression of genes associated with astrocytic fate (Clu, Id3, and Tubb4b) and a notable absence of neuronal phenotype and cell cycle genes that were present in NSC, IPC, NB, and IN clusters (Fig 5C).

We next identified differentially expressed genes (DEGs) between WT and iKD cells within clusters that contained NSCs and/or IPCs. Across all NSC or IPC containing clusters, 85 unique DEGs were identified (22 up in iKD and 63 down in iKD) (Table S1). Although Vegfa was detected in NSCs, IPCs, and astrocytes, expression levels were near the floor of detection in all cell types and therefore not fit for comparison between genotypes. To understand the potential functional significance of DEGs generated from this dataset, we used GO analysis of up-regulated and down-regulated DEGs coupled with kappa score iterative grouping with ClueGO (Bindea et al, 2009, 2013) to collapse similar GO terms. GO terms were associated with several processes that would be expected based on the role of NSPC-derived VEGF in maintaining stemness/self-renewal (e.g., differentiation processes), as well as processes consistent with VEGF signaling via VEGF receptor tyrosine kinases (e.g., serine/threonine kinase activities) (Fig 5D and Table S1) (Kirby et al, 2015). More relevant to the phenotype of disruption of vessel association after NSPC-VEGF loss, we found many differentially

regulated processes were associated with cell-substrate junction organization, a crucial part of cell anchoring and cell motility.

When down-regulated DEGs associated with cell-substrate junction were dissected by cluster, no strong pattern was noted by cluster; rather, DEGs appeared to be generally down-regulated throughout the NSC and IPC containing clusters (Fig 5E). Top down-regulated genes associated with cell–substrate junction included Hspa8, Pik3r1, Ptprj, Ptk2b, and Trem2. Hspa8 codes for a member of the heat shock protein 70 family and has established roles in promoting cell proliferation/survival (Calderwood et al, 2006; Ramírez-Rodríguez et al, 2013) as well as migration and invasion in cancer cells (Sun et al, 2019). Pik3r1 and Ptk2b both encode tyrosine kinases, whereas Ptprj codes for a tyrosine phosphatase. All three of these genes can promote both cell proliferation and migration (Jiang et al, 2020; de Pins et al, 2021; Li et al, 2022). Trem2 is a transmembrane receptor protein that is frequently associated with immune cells (e.g., microglia, macrophages) but is also known to be overexpressed in many cancer cells, where it can regulate both proliferation and tumor cell migration (Wolf et al, 2022). Only one DEG associated with cell–substrate junction (Enpp2) was up-regulated in iKD mice and it was similarly up-regulated throughout clusters (Fig 5E). Enpp2 codes for a soluble, secreted lysophospholipase D, also known as autotaxin, that can promote tumor progression because it can both promote motility of cancer cells in an autocrine manner as well as inhibit motility of immune cells, acting as a chemo-repellent (Borza et al, 2022). Taken together, our single-cell data suggested that loss of NSC-specific VEGF may disrupt regulation of cell adhesion/motility, in addition to its already expected role in regulation of proliferation/self-renewal. A disruption in NSC motility and adhesion presented a likely mechanism for loss of proximity to vessels after VEGF loss, which we next sought to verify.

## Inhibition of VEGFR2 inhibits NSC motility and adhesion in vitro

To confirm that NSC motility and adhesion rely on self-synthesized VEGF, we chose to focus on functional assays with cultured NSCs derived from adult DG. We chose functional assays of cell motility rather than focus on individual DEGs because previous research shows scRNAseq DEG analysis is highly subject to the bias of single-cell sequencing for detecting highly expressed genes with low fold changes (Squair et al, 2021; Denninger et al, 2022). We chose to focus on cultured NSCs because they provided a well-validated model of NSC physiology isolated from the confounding factors of interaction with other niche cell types. Our previous characterization of proliferating NSC cultures using immunolabeling, bulk RNAseq, and single-cell RNAseq revealed that they are comprised almost entirely of cycling and quiescent NSCs that show strong transcriptional similarity to in vivo RGL-NSCs, with very few committed progenitors present (Denninger et al, 2020, 2022). Others have similarly shown that these cultures can be effective tools for

---

clusters that included NSCs or IPCs, with WT and iKD cells shown as separate colors (left). GO biological process clusters that were differentially expressed in VEGF iKD NSC and IPC containing clusters (right). **(E)** Heat map of expression levels of top down-regulated genes associated with cell-substrate junction organization in WT and iKD NSC and IPC containing clusters. Values are log₂ fold change of the average normalized transcript count in a group over the cluster average.

modeling RGL-NSC physiology and function (Blomfield et al, 2019; Urbán et al, 2019; Petrelli et al, 2023).

To test the role of autocrine VEGF signaling in proliferative NSC motility and adhesion in vitro, we used a widely used, cell-permeable inhibitor of VEGFR2 receptor tyrosine kinase activity, SU5416. The defined media of standard NSC culture does not include any exogenous VEGF, making any VEGFR2 signaling dependent on autocrine VEGF signaling. We and others have previously reported that NSCs in vitro and in vivo express VEGFR2 but not the other major VEGF receptor, VEGFR1 (Fabel et al, 2003; Cao et al, 2004; Wittko et al, 2009; Kirby et al, 2015). As further confirmation that adult RGL-NSCs express VEGFR2, we re-examined the datasets from Berg et al (2019) and Adusumilli et al (2021), and we found that hippocampal NSCs express Kdr (the transcript for VEGFR2) despite a decline in expression with age (Fig S2A) and that Kdr expression is 524 ± 23% higher in DG than in SVZ NSPCs (Fig S2B). Co-labeling of brain sections with an antibody against VEGFR2 and an antibody against Nestin also showed prominent immunoreactivity for VEGFR2 in Nestin[+] NSCs (Fig 6A). Although some studies have questioned the efficacy of some VEGFR2 antibodies in fixed mouse tissue because they fail to show the expected vascular labeling pattern (Licht et al, 2016), the antibody used here was knockout validated for immunolabeling in mouse brain tissue (Harde et al, 2019) and also showed the expected pattern of VEGFR2 labeling in CD31[+] blood vessels (Fig S2C and D). VEGFR2 labeling was also evident overlapping with putative GFAP+ RGL processes (Fig S2E).

To test proliferative NSC motility, we used a scratch assay. We treated adult DG NSCs grown in standard proliferative monolayer conditions with SU5416 for 24 h before making a scratch through the monolayer (Fig 6B). We found that NSC ingression into the scratch was slowed in SU5416-treated NSCs compared with control NSCs (Fig 6C and D). To test proliferative NSC adhesion, we used an attachment assay (Yeo et al, 2023). After 24 h pretreatment with SU5416 (or vehicle DMSO), we treated NSCs in standard proliferative monolayer conditions with cell-permeable Hoechst dye to label nuclei then imaged the monolayers. After a brief treatment with the enzymatic cell detachment medium accutase, we then imaged again and quantified the percent of remaining cells (Fig 6E). SU5416 treatment reduced cell adhesion by ~31% compared with DMSO-treated controls (Fig 6F and G), suggesting reduced cell adhesion in the absence of VEGFR2 signaling.

The proliferative NSC model, as noted above, primarily reflects the cellular physiology of cycling NSCs. To assay motility and adhesion of more quiescent NSCs, we used the BMP4 model of acquired quiescence. When adult DG NSCs are denied exogenous EGF and instead supplemented with BMP4 for 3 d, they enter a preserved, reversible quiescent state (Blomfield et al, 2019; Urbán et al, 2019; Petrelli et al, 2023). We shifted adult DG NSCs to quiescent conditions then performed a scratch assay. Consistent with findings in SVZ NSCs (Yeo et al, 2023), quiescent DG NSCs were not motile; no migration into the scratch was observed up to 30 h after the scratch (Fig S2F). For the attachment assay, we used the detachment enzyme trypsin, as accutase proved unable to detach quiescent NSCs in our hands and others' (Yeo et al, 2023) (Fig 6H). 24 h pretreatment with SU5416 led to a ~25% reduction in cell adhesion after trypsin treatment compared with DMSO-treated controls (Fig 6I and J), again suggesting reduced cell adhesion in the absence of VEGFR2

signaling. Taken together, our in vitro results support our in vivo transcriptomic results, suggesting that NSC motility and adhesion relies on self-derived VEGF signaling.

## Discussion

Crosstalk between adult NSCs and their niche is essential for preservation of neurogenesis through adulthood. The relationship of NSCs and their progenitors with the unique vascular architecture of the adult DG is one of the most prevalent features of this niche. Here, we show that NSPCs are essential to preserving their own proximity to vessels through autocrine VEGF signaling. These findings reveal that the placement of NSPCs in their vascular niche is not fixed, but rather is actively maintained by NSPCs.

The hallmark features of the adult DG neurogenic vascular niche are as follows: (1) density of local blood vessels and (2) NSPC physical apposition to these vessels (Palmer et al, 2000; Sun et al, 2015; Moss et al, 2016; Licht et al, 2020). There is little information available about the molecular cues that support maintenance of these properties in adulthood. Prenatally, embryonic NSPCs promote de novo vascularization of the CNS via secretion of VEGF, which is a chemoattractant and mitogen for endothelial cells (Gerhardt et al, 2003; Haigh et al, 2003; Raab et al, 2004). VEGF expression in the adult brain is typically attributed to mature astrocytes (Bozoyan et al, 2012; Licht & Keshet, 2013; Kim et al, 2021). However, adult brain vasculature is frequently described as quiescent and VEGF-refractory, obtaining an independence from the reliance on VEGF signaling that characterizes developing vasculature (Wälchli et al, 2023). Our findings align with the concept of adult vasculature being VEGF-refractory. Although we observed a loss of NSPC-vessel association with global and NSPC-VEGF knockdown, we were unable to detect any changes in vascular architecture.

Although we did not observe changes in vascular structure of the DG niche, we observed prominent changes in the motility and cell adhesion of adult NSCs in vitro after loss of VEGF signaling, coupled with gene expression changes in cell adhesion processes in vivo. Changes in both cell adhesion and cell motility in NSPCs are likely drivers of disrupted maintenance of vascular proximity. Although motility and adhesion can be regulated independently, they are frequently intertwined, as migration can be promoted by adhesion to substrates but also inhibited if binding is too strong. DG IPCs are highly motile under normal conditions, adhering to the vasculature as a scaffold to migrate tangentially up to hundreds of microns as they differentiate into neuroblasts (Sun et al, 2015). In contrast, in vivo imaging reveals that the largely quiescent RGL-NSCs are less migratory (Pilz et al, 2018; Bottes et al, 2021; Wu et al, 2023). Our in vitro findings recapitulated these in vivo phenotypes, with NSCs in quiescent conditions failing to migrate into a scratch, whereas NSCs in proliferative conditions readily migrated into a scratch within hours.

The signals regulating the motility and adhesion properties of NSPCs are generally not clear. In the SVZ, quiescent NSCs are relatively stationary, much like DG RGL-NSCs, and their proliferative progeny are highly motile, like DG IPCs. A recent study showed that the motility and cell adhesion properties of these SVZ cells are

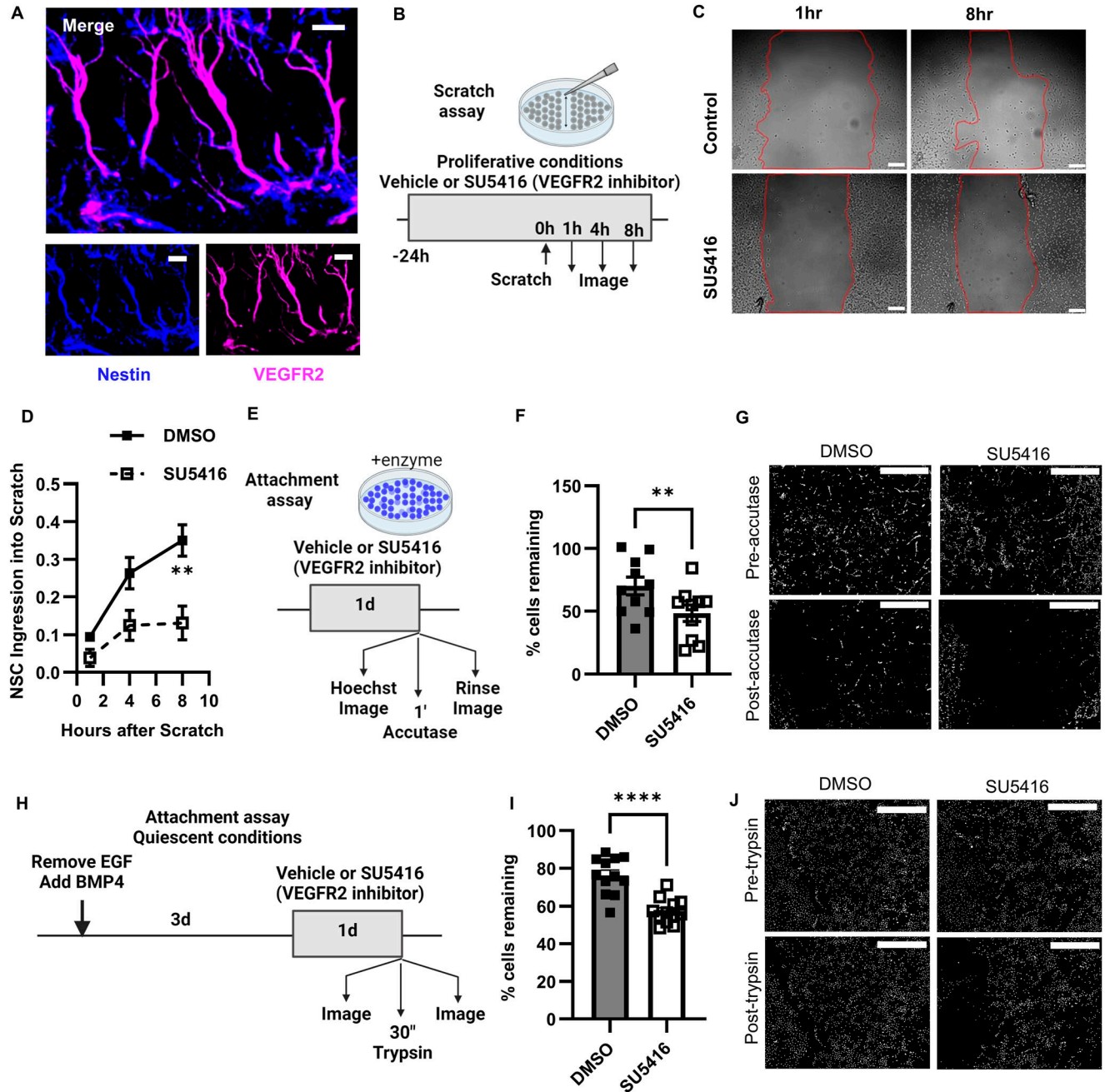

**Figure 6. NSC-VEGF regulates cell motility and attachment in vitro.**
**(A)** Representative immunofluorescent image of VEGFR2 immunolabeling with Nestin. **(B)** Diagram of scratch assay experimental design in proliferative conditions. **(C)** Representative bright field images of NSC migration into scratch after SU5416 treatment or control after 1 and 8 h. Red line = scratch border. **(D)** Comparison of NSC ingression into scratch. Data normalized to initial scratch size such that 0 is no ingression and 1 is complete closure of the scratch area. N = 3 wells/experiment, three experiments. Mean ± SEM shown. Post hoc comparison between SU5416 and veh wells within time point shown as **. **(E)** Diagram of attachment assay in proliferative conditions. **(F)** Percent of cells remaining after accutase treatment in veh or SU5416-treated NSCs in proliferative conditions. Points are individual wells. Mean ± SEM also shown. N = 2–4 wells/treatment/experiment, three experiments. Main effect of treatment across experiments shown as **. **(G)** Representative Hoechst+ cell nuclei before and after accutase in vehicle and SU5416-treated wells. **(H)** Diagram of attachment assay after transition to quiescent conditions. **(I)** Percent of cells remaining after trypsin treatment in veh or SU5416-treated NSCs in quiescent conditions. Points are individual wells. Mean ± SEM also shown. Main effect of treatment across experiments shown as **. **(J)** Representative Hoechst+ cell nuclei before and after trypsin in vehicle and SU5416-treated wells. N = 4 wells/treatment/experiment, three experiments. Scale bars represent (A, C) 100 μm, (G, J) 1 mm. *P < 0.05, **P < 0.01.

linked and can change with stimuli such as advancing age (Yeo et al, 2023). Our in vitro motility and adhesion assays, coupled with our in vivo transcriptomics, align to suggest that adhesion and motility are promoted by autocrine VEGF signaling in adult DG NSCs. Based on our data, we propose that NSPC-derived VEGF likely supports maintenance of NSPC vascular proximity and contact by regulating

NSPC motility, maintenance of cell adhesion, or both. Future research will be necessary to further dissect how NSPC-derived VEGF supports these processes.

The mature DG vascular niche is widely hypothesized to support adult neurogenesis, in particular via exposure to circulatory factors (Karakatsani et al, 2019). Beneficial blood-borne factors, such as those associated with exercise (Trejo et al, 2001; Fabel et al, 2003; Horowitz et al, 2020), as well as detrimental factors, such as those associated with old age (Villeda et al, 2011), drive changes in NSPC proliferation that result in altered neurogenesis and correlated improvements or impairments, respectively, in hippocampus-dependent memory function. The unique association of NSPCs with local capillaries is frequently cited as a likely mechanism for how circulating factors drive robust changes in NSPC proliferation and thereby neurogenesis (Licht et al, 2020; Kim et al, 2021). Our findings therefore highlight a new potential mechanism by which NSPC-derived VEGF could support maintenance of NSCs and thereby adult neurogenesis. Although the distance-to-vessel changes observed after loss of NSPC-VEGF in the present study were often small in magnitude (2–3 $\mu$m), this difference was enough to cause a complete loss of preferential IPC-vessel association beyond that found for a random SGZ cell. The functional relevance of loss of NSPC preferential association with vessels remains unclear, though. Although we previously showed that NSPC-VEGF loss impairs NSC maintenance (Kirby et al, 2015), it is important to note that with this model, we cannot discern whether that effect is due to loss of vascular proximity or loss of direct VEGF receptor signaling to support stemness independent of vascular signals. We also cannot be sure whether any of the changes in relationship to vasculature here are a cause of loss of stemness or are a consequence of it.

There are several limitations to the present study. First, although we have previously shown VEGF protein production by NSPCs as a population (Kirby et al, 2015), our analysis of VEGF expression in single RGL-NSCs here relies mostly on transcriptional-level analysis. Identification of the cellular source of VEGF protein is complicated by the fact that secreted proteins have short intracellular half-lives and most immunolabeling for VEGF is therefore found in the interstitial space or bound to/inside receptor-expressing cells. In addition, RNAscope analysis would be more certain if additional markers to differentiate astrocytes from NSCs were used. Second, although we did not detect changes to the vasculature, our subregion analysis may have been too broad to detect subtle changes in vessel location. Additional studies could use live imaging after NSPC-VEGF loss to determine if there is more subtle endothelial migration that contributes to loss of NSPC-vessel proximity. In the present work, we used distance from cell center plus assessment of whether RGL-NSC processes appeared to intersect with a blood vessel as measures of vascular association. Neither of these measures takes into appreciation the full 3D shape of the cells and how they contact vessels. 3D imaging of thick, cleared tissue samples may provide a more nuanced view of vascular structure and NSPC relationship to that vasculature. Third, although we saw no changes in occludin expression with VEGF iKD, these findings do not comprehensively assess changes to BBB permeability. However, VEGF is a permeabilizing factor for the BBB and the BBB in the DG is already reported to be intact (Shen et al, 2004), making it unlikely that loss of VEGF could affect much change under normal circumstances. There is still a possibility that BBB permeability induced by surges of VEGF after an injury (e.g., seizures. stroke) may depend on NSPC-produced VEGF. This possibility remains a topic for future research. Finally, more work is needed to better define the downstream mediators of VEGF signaling in adult DG NSCs. Our single-cell sequencing data provide a large dataset that implicates several pathways as dependent on NSPC-VEGF signaling. Future research will be necessary to determine the functional roles of those signaling events disrupted downstream of VEGF loss.

In conclusion, determining the key mechanisms by which adult DG NSCs regulate themselves and their microenvironment is an important step in understanding the preservation of the adult hippocampal neurogenic niche. Our findings suggest that autocrine NSPC-derived VEGF maintains NSPC proximity to blood vessels. Future studies may examine changes to bidirectional signaling between NSPCs and endothelia to determine if loss of VEGF changes endothelial gene expression, resulting in a loss of migratory or adhesive signals for NSCs. They may also investigate how different sources of VEGF or manipulations to VEGFR2 expression may preserve neurogenic capacity, for example, to protect against loss of neurogenesis with aging or injury. These findings have important implications when considering how to therapeutically support endogenous adult neurogenesis or exogenous NSC transplants.

# Materials and Methods

### Mice

All animal use was in accordance with institutional guidelines approved by The Ohio State University Institutional Animal Care and Use Committee. All mice were 7–10 wk old at the start of any experimental manipulation. NestinCreER[T2] mice (#016261; Jackson) were crossed with Rosa-stop-floxed-EYFP ([Srinivas et al, 2001]; #006148; Jackson) and *Vegfa*[lox] mice, provided by Genentech, Inc (Gerber et al, 1999), to create NSPC-specific VEGF knockdown for scRNAseq. Mice were maintained as NestinCreER[T2+/−];*Vegfa*[lox/wt];Rosa-EYFP[+/+] x NestinCreER[T2−/−];*Vegfa*[lox/wt];Rosa-LoxSTOPLox-EYFP[+/+] breeding pairs. NestinCreER[T2+/−];*Vegfa*[wt/wt];Rosa-EYFP[+/+] (WT) and NestinCreER[T2+/−];*Vegfa*[lox/lox];Rosa-LoxSTOPLox-EYFP[+/+] (iKD) littermates were used for scRNAseq. Wild type C57BL/6J male and female mice were purchased from Jackson Laboratory (#000664; Strain). VEGF-GFP mice were a gift from Brian Seed, Harvard University, Cambridge, MA, USA (Fukumura et al, 1998). All mice were housed in standard ventilated cages, with ad libitum access to food and water throughout all experiments and maintained on a 12-h light cycle with lights on at 630 h. Male and female mice were represented in approximately equal numbers throughout and no differences in sex were found. We therefore combined them in all presented analysis.

### Tamoxifen administration

Tamoxifen (TAM; APExBIO) was dissolved in sterile sunflower oil at 20 mg/ml, overnight with agitation at 37°C and stored at +4°C for up to 1 wk in the dark. TAM was injected (180 mg/kg/d, i.p.) for 5 d.

## NSC isolation

NSCs were isolated from adult hippocampus of C57BL/6J mice as described in Babu et al (2011). Unless otherwise stated, NSCs were maintained on poly-D-lysine (Sigma-Aldrich) and laminin (Invitrogen) coated plates in Neurobasal A medium (Invitrogen) with 1x B27 supplement without vitamin A (GIBCO), 1x GlutaMax (Invitrogen) and 20 ng/ml each of EGF and FGF2 (Peprotech), as per (Babu et al, 2011). No cells were used past passage 20. Two separate lines were used in all experiments, one from five pooled C57BL/6J male mice and one from five pooled C57BL/6J female mice. No differences between NSCs isolated from males and females were found.

## RNAscope in situ hybridization and immunohistochemistry

WT C57BL/6J mice were transcardially perfused with ice-cold PBS followed by 4% PFA. Brains were harvested and fixed overnight at 4°C in 4% PFA before serial overnight equilibration in 10%, 20%, and 30% sucrose. Fixed tissue was snap frozen in OCT in a dry ice/100% ethanol bath and stored at −70°C. 12 $\mu$m cryosections, one section per slide, were prepared with a cryostat and stored at −70°C with desiccant until staining. RNA in situ hybridization was performed according to manufacturer recommendations for using fixed frozen tissue samples in the RNAscope Multiplex Fluorescent v2 Assay (Advanced Cell Diagnostics) with the following modifications to enable concurrent immunohistochemical staining. The pretreatment steps were replaced with a 15 min modified citrate buffer (Dako) antigen retrieval step in a steamer at 95°C. In addition, the protease III step was excluded to enable subsequent immunohistochemical staining. Probes for mouse Vegfa (Mm-Vegfa-ver2; ACD) RNA were hybridized to tissue before subsequent immunohistochemical staining for GFAP protein. Immunostaining for GFAP was conducted as described in (Immunohistochemical tissue processing) with the following exceptions. Blocking was performed with 10% normal donkey serum in 0.1 M TBS-1% BSA. Antibody incubations were performed in TBS-1% BSA. All washes were performed with TBST. DAPI provided by the RNAscope Multiplex Fluorescent kit was used for nuclear counterstaining. All images were acquired with the Zeiss Axio Observer Z1 microscope with Apotome for optical sectioning using a 20x air objective. Full z-stacks were acquired for analysis.

## RNAscope analysis

RGL-NSCs and astrocytes were identified based on the morphology of GFAP$^+$ apical or stellate processes, respectively, extending from a Hoechst$^+$ nucleus in 1 $\mu$m z-stack images from n = 4 mice. Vegfa puncta were found almost exclusively in the nucleus in both cell types and were counted manually throughout the depth of the nucleus.

## FACS for 10x genomics single-cell sequencing and analysis

NestinCreER$^{T2+/−}$; $Vegfa^{lox/lox}$; ROSA-LoxSTOPLox-EYFP$^{+/+}$ (KD; n = 5) and NestinCreER$^{T2+/−}$; $Vegfa^{wt/wt}$; ROSA-LoxSTOPLox-EYFP$^{+/+}$ (WT; n = 5) littermates were submitted to tamoxifen injections (described above) and 3 wk later their DGs were dissected, as described above.

DGs were then mechanically dissociated with sterile scalpel blades then repeatedly with mortar and pestle in douncing buffer on ice. Dissociated cells were collected by centrifugation at 500$g$ for 5 min. Cells were then in an HBSS without Ca$^{2+}$/Mg$^{2+}$/Percol solution to form a gradient and spun at 450$g$ for 15 min at room temperature. The supernatant was removed, and the remaining cells were resuspended in a solution containing HBSS with 0.04% BSA. Cells were then filtered through a 35 $\mu$m nylon filter before staining with Hoechst dye on ice for 10 min for live/dead discrimination. All cells were washed twice following staining and immediately sorted on the FACS Aria III (BD Biosciences) where EYFP$^+$ cells were collected for scRNAseq in the HBSS 0.04% BSA buffer. Of those, ~10,000 cells per genotype were loaded onto the 10× Genomics single-cell sequencing platform using the standard kit. The 3' RNAseq library was sequenced using paired-end 150 bp approach on an Illumina NovaSeq 6000 sequencer. CellRanger 7.0.1 (Zheng et al, 2017) was used to demultiplex, align, and deduplicate sequencing reads in BCL files. Single-cell data in feature-barcode matrices were then processed using Seurat v4.1.1's default pipeline (Hao et al, 2021) to identify unsupervised cell clusters and generate a UMAP plot. In brief, cells were filtered to exclude multiplets and damaged cells by excluding cells with unique feature count <2,000 or >5,000 and a mitochondrial gene expression of >2.0%. From ~10,000 cells loaded per genotype, ~5,500 were recovered in both genotypes, yielding a net capture rate of 55%. Data were then log normalized with default scale factor of 10,000 and integrated into one file containing both KD and WT samples. The data underwent linear transformation (ScaleData function) and PCA was run on the scaled data, followed by FindNeighbors and FindClusters. UMAPs were then created using the RunUMAP function. DEGs defining clusters (regardless of treatment) and defining treatments (within cluster) were generated using the FindAllMarkers function, which uses a default of Wilcoxon rank sum test, unless otherwise noted. Adjusted $P$-values are Bonferroni-corrected using all features in the dataset. DEGs were further analyzed in ClueGO/Cytoscape (Shannon et al, 2003; Bindea et al, 2009, 2013) using biological process for kappa score iterative grouping.

## Lentiviral vector production

VEGF and scramble shRNA lentiviral vectors are described by Mosher et al (2012) and were packaged in VSV-G lentivirus by either Vigene Biosciences or the Stanford Gene Vector and Virus Core.

## Scratch assay

NSCs were plated into a 24-well plate coated with laminin and PDL at a density of 50,000 cells per well. SU5416 was dissolved in DMSO (10 mM) and stored at −20°C until use. SU5416 was used at 25 $\mu$M. For proliferative condition experiments, after 24 h in SU5416/vehicle, a scratch was performed with a sterile pipet tip. The scratch was then imaged at 0, 1, 4, and 8 h to observe NSC migration, while incubating at 37°C between the time points. Scratch area was determined at each time point in Zen software. Because each scratch was unique, areas for each scratch were individually normalized to the area at 0 h. Scratch ingression was calculated as 1- (scratch area at hour X/ scratch area at 0 h). On this scale, 0 represents no ingression and 1

**Table 1.  Materials and reagents.**

| Reagent or resource | Source | Identifier |
|---|---|---|
| **Antibodies** | | |
| Hoechst 33342 (IF, attachment assay; 1:2,000) (Flow; 1: 5,000) | Thermo Fisher Scientific | Cat#H3570 |
| Anti-Glial Fibrillary Acidic Protein, Clone: GA5 (IF; 1:1,000) | Sigma-Aldrich | Cat#MAB360 RRID: AB_11212597 |
| Anti-GFP Antibody (IF; 1:1,000) | Abcam | Cat#ab6673 RRID: AB_305643 |
| BrdU Antibody (IF; 1:500) | Bio-Rad | Cat#MCA6114 RRID: |
| SOX2 Rat anti-Human, Mouse, Clone: Btjce (IF; 1:1,000) | eBioscience | Cat#50-112-9095 RRID: |
| Mouse anti-Nestin (IF, 1:100) | Millipore | Cat#MAB353 RRID: AB_94911 |
| Goat anti-VEGFR2 (IF in vivo, 1:100) | R&D Systems | Cat#AF644 RRID: AB_355500 |
| Jackson Immuno Research Labs Biotin-SP (long spacer) AffiniPure Donkey Anti-Rabbit IgG (H+L) | Thermo Fisher Scientific | Cat#NC9745676 RRID: AB_2340593 |
| Donkey anti-Goat IgG (H+L) Cross-Adsorbed Secondary Antibody, Alexa Fluor 488 (IF; 1:500) | Thermo Fisher Scientific | Cat#A-11055 RRID: AB_2534102 |
| Donkey anti-Rabbit IgG (H+L) Highly Cross-Adsorbed Secondary Antibody, Alexa Fluor 555 (IF; 1:500) | Thermo Fisher Scientific | Cat#A-31572 RRID: AB_162543 |
| Donkey anti-Goat IgG (H+L) Cross-Adsorbed Secondary Antibody, Alexa Fluor 555 (IF; 1:500) | Thermo Fisher Scientific | Cat#A-21432 RRID: AB_2535853 |
| Donkey anti-Mouse IgG (H+L) Highly Cross-Adsorbed Secondary Antibody, Alexa Fluor 647 (IF; 1:500) | Thermo Fisher Scientific | Cat#A-21447 RRID: AB_141844 |
| Donkey anti-Rat IgG (H+L) Highly Cross-Adsorbed Secondary Antibody, Alexa Fluor 594 (IF; 1:500) | Thermo Fisher Scientific | Cat#A-21209 RRID: AB_2535795 |
| IgG (H+L) Highly Cross-Adsorbed Donkey anti-Rabbit, Alexa Fluor 350 (IF; 1:500) | Thermo Fisher Scientific | Cat#A-10039 RRID: AB_2534015 |
| **Chemicals, peptides, and recombinant proteins** | | |
| SU5416 (25 μM) | Sigma-Aldrich | Cat#S8442 |
| Animal-Free Recombinant Human EGF, PeproTech (AF10015-500UG) | VWR | Cat#10781-694 |
| PeproTech Recombinant Human FGF-Basic (154A.A) (100-18B-250UG) | VWR | Cat#10771-938 |
| TRITON X-100 | Thermo Fisher Scientific | Cat#21568 |
| 5-bromo-2'-deoxyuridine | Sigma-Aldrich | Cat#B5002 |
| 5-Ethynyl-2'-deoxyuridine | Click Chemistry Tool Kit | Cat#1324 |
| **Critical commercial assays** | | |
| RNAscope Multiplex Fluorescent Assay v2 | ACD Biotechne | Cat#323136 |
| RNAscope probe Mm-*Kdr*-C2 | ACD Biotechne | Cat#414811-C2 |
| Mouse VEGF DuoSet ELISA | R&D Systems | Cat#DY493-05 |
| **Deposited data** | | |
| Single-Cell RNAseq Data Set | | GEO#: GSE220871 |
| **Experimental models: Cell lines** | | |
| Cultured Neural Stem Cells | | Babu et al (2011) |
| **Experimental models: Mouse Models** | | |
| Wild type C57BL/6J | Jackson | #000664 |
| *Vegfa*^lox/lox | | Gerber et al (1999) |
| VEGF-GFP | | Fukumura et al (1998) |
| Rosa-stop-floxed-EYFP | Jackson | #006148; Jackson |
| NestinCreER^T2 | Jackson | #016261; Jackson |

**Table 1.  Continued**

| Reagent or resource | Source | Identifier |
|---|---|---|
| **Experimental models: Lentiviral Vectors** | | |
| GFP$^+$ scramble shRNA | Mosher et al (2012) | |
| GFP$^+$ Vegfa shRNA | Mosher et al (2012) | |
| **Software and algorithms** | | |
| ImageJ | schneider | https://imagej.nih.gov/ij/ |
| Zen | Zeiss | https://www.zeiss.com/microscopy/en/products/software/zeiss-zen.html |
| Cell Ranger Cloud Analysis 7.0.1 | 10x Genomics | Zheng et al (2017) |
| Seurat 4.1.1 | Satija Lab | Hao et al (2021) |
| Cytoscape 3.9.1 | | Shannon et al (2003) |
| ClueGO | | Bindea et al (2013 and 2009) |
| Prism | GraphPad | https://www.graphpad.com/scientific-software/prism/ |

represents complete scratch closure. For quiescent conditions, NSCs were plated as above initially in proliferative conditions. 24 h later, the medium was changed to growth medium with no EGF and 20 ng/ml BMP4 (FGF2 remained at 20 ng/ml). 4 d after medium change, a scratch was performed as above and imaged 0, 1, 4, 8, and 30 h later.

### Detachment assay

For all experiments, NSCs were plated in 96-well plates in a single row of eight wells at a density of 5,000 cells per well. For proliferative condition experiments, 24 h after plating, alternating wells were treated with 25 $\mu$M SU5416 or DMSO vehicle. 24 h after treatment, Hoechst 33342 was spiked into each well at 1:2,000 final dilution and incubated for 10 min at 37°C, 5% $CO_2$. Cells were then imaged at 10x magnification in a 3 × 3 grid. The medium was dumped from the whole plate at once and accutase was added with a multichannel pipettor to assure similar physical forces on all wells. Accutase was dumped out of the plate after 1 min. A single rinse in Neurobasal A with Hoechst 1:2,000 was performed with a multichannel pipettor and cells were left in Neurobasal A with Hoechst 1:2,000 for immediate imaging. Imaging was performed in the same location using digital anchors placed in Zen software during the initial imaging. For quiescent conditions, the procedure was similar, except cells were switched to quiescence medium (no EGF, +BMP4 20 ng/ml) for 3 d before SU5416 or vehicle was added. 24 h after treatment, 30 s of trypsin was used instead of accutase as the detachment reagent because accutase was not sufficiently stringent to result in any detachment of quiescent NSCs.

Hoechst+ cells were counted using Image J automated cell counting on thresholded images (analyze particles function). The number of cells in a well after detachment was divided by the number of cells in that well before detachment to yield a % retained. For statistical analysis, wells were considered in pairs that were adjacent on the same plate, to help match physical forces experienced during dumping and medium changes.

### Stereotaxic surgery

Mice were anesthetized by inhalation of isoflurane (5% induction, 1–2% maintenance; Akorn) in oxygen and mounted in the stereotaxic apparatus (Stoelting). Ocular lubricant (Puralube) was placed over the eyes to prevent evaporative dry eye. After sterilization with alcohol (Thermo Fisher Scientific) and betadine swabs (Thermo Fisher Scientific), the skull was exposed and the lambda and bregma sutures were aligned in the same horizontal plane. A small bur hole was drilled in the skull and an automated injector (Stoelting) with a Hamilton syringe (Hamilton) was lowered to the injection depth at a rate of –1.0 mm/min. Mice were injected with 0.5 $\mu$l of scramble control shRNA virus into one hemisphere and 0.5 $\mu$l of Vegfa shRNA virus into the contralateral hemisphere at a rate of 0.1 $\mu$l/min. The injection coordinates from bregma were as follows: anterior/posterior –1.9 mm, medial/lateral ±1.6 mm, and –1.9 mm dorsal/ventral from dura. Post-surgery, the incision was sealed with tissue adhesive (3M) and the mouse was given saline (Hospira) and carprofen (Zoetis) injection i.p. After 21 d, mice were perfused for immunohistochemical processing.

### EdU and BrdU labeling

21 d after TAM administration, mice were injected with 5-bromo-2′-deoxyuridine (BrdU, 150 mg/kg, IP) (Sigma-Aldrich) and euthanized 2 h later for tissue processing. In virus-treated mice, after 21 d, mice were injected with 5-Ethynyl-2′-deoxyuridine (EdU) (Click Chemistry Tools) dissolved in physiological saline (Hospira) (150 mg/kg, IP) and euthanized 2 h later for tissue processing.

### Immunohistochemical processing

Brains for immunolabeling were harvested after perfusion with ice-cold PBS followed by fixation in 4% PFA overnight at 4°C. After equilibration in 30% sucrose in PBS, 40-$\mu$m coronal brain sections were obtained in 1 in 12 series on a freezing microtome (Leica) and

stored in cryoprotectant at −20°C. Sections were rinsed 3x in PBS and incubated in a blocking solution containing 1% normal donkey serum and 0.3% Triton X-100 (Acros) in PBS before incubation in primary antibodies for 24–72 h. Sections were rinsed 3x with PBS and exposed to a secondary antibody diluted in blocking solution for 2 h. For VEGF-GFP co-labeling with nestin, sections were incubated with a biotinylated secondary for GFP labeling followed by a streptavidin Alexa Fluor 488 tertiary (1:1,000 in PBS). BrdU immunolabeling was performed after other immunolabeling was complete. Labeled sections were fixed in 4% PFA for 10 min at room temperature, rinsed, and then incubated with 2N HCl for 30 min at 37°C. Sections were then rinsed, blocked, and incubated in anti-BrdU primary and appropriate secondaries as above. The DG of the hippocampus was imaged in 1 $\mu$m Z-stacks at 20x magnification using a Zeiss apotome digital imaging system (Zeiss).

### Immunofluorescent image quantification

In TAM-treated mice applications, RGLs were identified by GFAP$^+$/EYFP$^+$ colocalization and GFAP$^+$ radial processes extending from the SGZ towards the inner molecular layer, whereas MCM2$^+$/EYFP$^+$ cells in the SGZ layer were identified as IPCs. BrdU$^+$ nuclei were counted in the SGZ and granular cell layer. Neuroblasts were identified as DCX$^+$ cells in the SGZ with a bipolar morphology. Immature neurons were identified as DCX$^+$ cells with a primary dendrite extending through the granular cell layer. Endothelia were identified by CD31$^+$. Distance to vasculature was measured as the distance from the middle of a cell body (a hoescht+ nucleus) to the nearest CD31$^+$ vessel. Vessels were defined as elongated structures, distinct from punctate background. Random distances for vessel associations were measured by sampling the distance of random Hoechst$^+$ cells in the middle of the SGZ to the vasculature, spaced every ~100 $\mu$m along the SGZ midline. The SGZ was defined as the zone spanning 2 cell body widths between the dense granular cell layer and the hilus. CD31$^+$ cell density and Occludin/CD31 overlap were performed using thresholded area in ImageJ by a blind observer.

In other applications, NSCs were identified by GFAP$^+$/SOX2$^+$ colocalization and GFAP$^+$ radial processes extending from the SGZ towards the inner molecular layer, whereas GFAP$^-$/SOX2$^+$ cells in the SGZ layer were identified as IPCs. For RNAscope only, GFAP$^+$ apical process alone was used (without SOX2) because of limitations in the number of separate fluorescent wavelengths available and effectiveness of immunolabeling in combination with RNAscope. Cell counts were performed manually in Zen by a blind observer.

### Diagram creation

Cartoon diagrams in this article were created with BioRender.com.

### Statistical analysis

Statistical analyses were performed as described in Table S2 (except scRNAseq which is described above). In general, if pairs of groups were compared, $t$ tests were used. If more than two groups were compared with one factor, one-way ANOVA was used followed

by error-corrected post hoc tests. If groups with two factors were compared, two-way ANOVA was used followed by error-corrected post hoc tests. All analyses were performed using Prism (v9.0; GraphPad Software) and $P < 0.05$ was considered significant.

Product numbers and vendor details are available in Table 1.

## Data Availability

Single-cell RNAseq data are available at GEO accession GSE220871.

## Supplementary Information

## Acknowledgements

This work was funded by National Institutes of Health grant R01 NS124775 (ED Kirby), Ohio State College of Arts & Sciences Undergraduate Research Scholarship (R Osap).

### Author Contributions

TJ Dause: conceptualization, investigation, visualization, methodology, and writing—original draft, review, and editing.
JK Denninger: investigation, methodology, and writing—review and editing.
R Osap: investigation and writing—review and editing.
AE Walters: investigation and writing—review and editing.
JD Rieskamp: investigation, methodology, and writing—review and editing.
ED Kirby: conceptualization, supervision, investigation, visualization, methodology, and writing—original draft, review, and editing.

### Conflict of Interest Statement

The authors declare that they have no conflict of interest.

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
