## [Reviewer comments · Life Science Alliance]

Life Science Alliance

Autocrine VEGF drives neural stem cell proximity to the adult hippocampus vascular niche

Tyler Dause, Jiyeon Denninger, Robert Osap, Ashley Walters, Joshua Rieskamp, and Elizabeth Kirby

DOI: <https://doi.org/10.26508/lsa.202402659>

Corresponding author(s): Elizabeth Kirby, The Ohio State University

Review Timeline:

Submission Date:	2024-02-15
Editorial Decision:	2024-02-19
Revision Received:	2024-04-03
Editorial Decision:	2024-04-04
Revision Received:	2024-04-05
Accepted:	2024-04-08

Transaction Report:

Please note that the manuscript was previously reviewed at another journal and the reports were taken into account in the decision-making process at *Life Science Alliance*.

Reviews

Referee #1 Review

General comment

Neural stem cells (NSCs) and their progeny (IPC), (all together NSPCs), primarily locate in the subventricular zone (SVZ) and the dentate gyrus (DG) of the hippocampus. The vasculature is a key component of adult brain NSCs niches. NSCs reside in close contact with a dense capillary network in the adult mammalian hippocampus. Crosstalk between adult NSCs and the vasculature within its niche is essential for preservation of neurogenesis through adulthood. Therefore, understanding the communication between NSCs and the nearby endothelium may shed light onto how stemness is maintain and modulated within CNS neurogenic niches, which has obvious therapeutic implications for brain repair.

In this Manuscript, through a combination of in vivo experiments in mouse transgenic lines, biochemistry and in vitro work, Dause et al propose a model where intracrine VEGF-VEGFR2 signaling in NSCs is required to keep them close to blood vessels. To support this hypothesis the authors firstly used a VEGF-GFP mouse model to validate the expression of VEGF in NSPCs. Then, through ablation of VEGF in NSPCs (using a NesCreERT2; Vegflox/lox mouse), the authors found that NSPC proximity to vessels is disrupted upon NSPC-VEGF loss, apparently by NSPCs undergoing defective migration and impaired cell adhesion. The authors also indicate that VEGF receptors (VEGFR2) are not exposed in the plasma membrane of VEGF-sensitive NSPCs, but instead they exist in intracellular pools. Based in these findings the authors conclude that NSPCs regulate their proximity to the vasculature, and thus their stemness, through cell-autonomous regulation by an intracrine VEGF/VEGFR2/Akt-dependent signalling.

Although the idea is interesting, the two major findings of this work (namely disruption of NSPC proximity to vessels upon VEGF loss and intracrine VEGF/VEGFR2/Akt signalling), are not convincingly supported by the experimental data presented in the study.

Major concerns:

Recombination Efficiency in the iKD Model

> Neither in Figure 2A, E, F and H (VEGFlox/lox recombination) nor later in Figure 8B (lenti-based VEGF knockdown) did the authors prove the actual reduction of VEGF expression in NSPCs. This is specially important for interpreting the following data:

> Graphs in Figures 2C, 2D, and 2G: The data points representing the distance of RGL-NSCs and IPCs to the nearest vessels shows that the distribution of the majority of the iKD samples is similar as the WT ones. It is the case that only 4 iKD points (out of the 12 analyzed) appear to be the ones driving the statistical significance in this analysis. Therefore, if recombination efficiency was the same in all samples, this brings forward doubts on the relevance/strength of the authors claim.

> Although the authors had their reasons to attempt the shRNA-based knockdown of VEGF using lentivirus, the achieved knockdown of VEGF is global (of note: direct proof of VEGF knockdown in figure 8 is missing) affecting different cell types. The influence of global VEGF removal on the final NSPC-vessel distance should not be ruled out or neglected.

No experimental data to support the proposed molecular mechanism of VEGF secretion and intracellular signalling

> By showing that VEGFR2 is not expressed in the plasma membrane of VEGF-sensitive NSPCs but intracellularly instead, the authors propose a model - where all molecular steps are indicated in their model figure - where NSPCs can autonomously activate an intracrine VEGF/VEGFR2/Akt signalling to modulate their distance and association with vessels. Important aspects of the diagram in Figure 10 imply mechanisms that were not experimentally confirmed in the present study.

> Crucial to this intracrine signalling, is the removal of plasma membrane-exposed VEGFR2 by shedases. However, the only piece of the proposed mechanism that is experimentally addressed in this work is the involvement of Akt downstream of VEGFR2 activation. The authors do not explore any aspect of the subcellular localization or sorting of either VEGF or VEGFR2, or how do they come together intracellularly for the activation of the signal. Similar situation comes to the regulation of the sheddases that ultimately allow to favor intracrine vs.

autocrine/paracrine signalling. Moreover, no molecular mechanism is proposed for the apparent deregulation of migration and cell adhesion proposed to be the driver of the observed phenotype.

Points requiring further clarification or improvement

> The technical approaches to support some of the arguments are not clear.

> In Figure 1F, RGL-NSCs are differentiated from astrocytes based on their morphologies to measure the Vegfa expression in both cell types. An inclusion of astrocyte markers such as Sox9 could have better differentiated both cell types.

> The distance between CD31+ vasculature and RGL-NSCs was measured while considering the cell body center of EYFP+/GFAP+RGL-NSCs (Figure 2C). The changes in cell shape could bias this distance. A more reliable approach could be to measure the distance of the apical processes of NSCs from blood vessels.

> The authors have shown CD31 positive cells as individual dots instead of an intact vascular structure (Figure 2J). Similarly, in Figure 3E, CD31 staining is inconsistent in both wild type and iKD. Thus, conclusions drawn just from these figures are questionable.

> In Figure 4B, the cellular population from the wild type shows almost no astrocytes. If this is true, how is astrocyte-associated gene analysis from wild type compared with iKD?

> Fig 4E: The information intended to be displayed in the heat map should be further explained. It has all the cell types separately for WT and iKD, but the heatmap scale says "log2(Foldchange)" - fold change of what over what?

> Figure 8C shows no statistical significance between scramble and Vegfa lentiviruses vectors. Does it show the inefficiency of Vegfa shRNA in removing Vegfa significantly from GFP+ NSCs? Moreover, upon Vegfa treatment, GFAP projection is also altered upon VEGF depletion in RGL-NSCs (Fig 8E).

Referee #2 Review

In this manuscript Dause et al. address the role of VEGF in the maintenance and positioning of adult neural stem cells relative to the vasculature. This is an extension of previous work by the senior author showing that VEGF plays a role in the maintenance of hippocampal neuronal stem cells. The authors used a combination of conditional gene ablation of VEGF and in vitro experiments and claim that VEGF plays a role in maintaining neural stem cells in close proximity to the vasculature but works through a cell-autonomous mechanisms rather than through paracrine signalling. Although the question of whether VEGF plays a role on hippocampal neural stem cell maintenance and differentiation is not novel, the authors propose that loss of VEGF affects an intracellular autocrine signal that results in putative neural stem cells moving away from the peri-niche vasculature in the subgranular layer of the dentate gyrus. The authors generated considerable data quantifying changes in location of GFAP+ putative radial stem cells in the dentate gyrus relative to vascular cells. They then go on to perform single cell sequencing of cells isolated from the dentate gyrus of Vegfa wildtype and Vegfa conditional knockout mice. They identify changes in gene expression associated with cell adhesion and cell attachment following Vegfa ablation. They then test the role of VEGF in neural progenitor migration in scratch assays in vitro using siRNA knockdown approaches. They show that VEGF seems to work through VEGFR2 in an intracellular signalling pathway and that VEGFR2 is not expressed at the cell surface on hippocampal stem cells and seems to be released from the surface by a metalloprotease. The analysis is interesting but mainly descriptive and does not mechanistically go beyond the previous indication that VEGF could regulate neural stem cell maintenance. There are no really insights into the intracellular signalling pathway activated by VEGF and how this regulates neural stem cell behaviour. It is unclear which metalloprotease potentially regulates VEGFR shedding and whether this has a biological function in vivo.

Major concerns

The in vivo "location" defects detected in the VEGF conditional knockout mouse are rather modest. The cells move 2-3µm further away from the blood vessels (~14.5µm for control neural stem cells and 17µm for Vegfa conditional deleted cells). It is not clear and not addressed whether this movement is important for the changes in proliferation

and the maintenance defects caused by Vegfa deletion.

In Figure 2I the author show loss of putative stem cell associate with blood vessels. However, it is not clear what percent RGL processes contacting vasculature really means and whether this is relevant. It is also unclear if this phenotype is a cause of neural stem cell defects or as a result of changes in the neural stem cell maintenance as a result of Vegfa deletion.

Then authors show that loss of VEGF expression by Nestin::CreERT2 expressing cells and their progeny does not affect the vascular structure in the dentate gyrus but the graphs in Figure 3D and 3 F have statistics which is confusing. This just reflects vascular density differences across the different regions of the dentate gyrus. These data do not bring much to the main manuscript.

The single cell data analysis shows relatively minor effects on gene expression. In Figure 4C it is not clear what is meant by Average Expression - is this a Z-score? In addition, the fold changes shown in the heat map in Figure 4E are very low (Log2 fold change of +/- 0.1). Again the description of the analysis and quantification are not clear and I assume this is a Z-score of the fold change across all cell types. How many samples were analyzed here and how many cells are contribute to the gene expression heat map in each category?

Figure 5: The scratch assay is interesting but the relevance of this for neural stem cells that are sedentary in vivo is not clear. The labels of the axes of the graphs (in addition in most of the figures) is not clear and the authors need to explain what is exactly measured and the units of the y-axis'. Why does the control virus affect the migration of infected cells?

The images in Figure 5B and 5 D are not visible.

The authors should show if the surface sheading of VEGFR is required in vivo.

In Figure 7F, the authors need to explain why TAPI treatment affects AKT activity in the absence of VEGF if the assay should report the intracellular VEGF activity? Does TAPI treatment increase the AKT levels in response to VEGF treatment higher than the VEGF treatment alone?

Why are the distances of stem cells from the nearest vessel different in the controls in Figure 7C compared to the values in Figure 2C? This is particularly true for the Vegfa siRNA GFP- cells which should be wildtype or not?

Referee #3 Review

In their manuscript «Intracrine signaling drives neural stem cell proximity to the adult hippocampus vascular niche» Dause and colleagues aimed to characterize the mechanisms how neural stem cell (NSC)-derived VEGF affects neurogenesis in the adult mouse hippocampus. The work presented is a continuation of previous work from Kirby et al., 2015 (PNAS) that had described the expression of VEGF in hippocampal NSCs and the functional consequences of NSC-specific VEGF genetic deletion.

The experimental design of the current study is straightforward, the data are convincing. However, the advance provided is unfortunately somewhat limited.

There are clearly more in-depth analyses compared to the Kirby et al 2015 study but those new data are largely "details". Indeed, the 2015 PNAS used similar approaches (e.g., VEGF-GFP mice) to analyze expression of VEGF (Kirby Figure 1 is somewhat similar to the current Figure 1), genetic deletion of VEGF was done using the "same" NestinCreERT2 mice in the current and previous study, vascular density was analyzed in the 2015 study using CD31 densities (as it was done in the current study). There is no question that the current ms contains novel data (e.g., distances from vessels of NSCs; even though this may be also influenced by the increase of proliferative cells upon VEGF-deletion) and subsequent molecular analyses, e.g., using transcriptomics. However, the interpretation that the intracrine loop (it is not entirely clear why the authors suggest that the VEGF loop is intracrine rather than autocrine; given their 2015 data suggesting that NSCs secrete VEGF in substantial amounts) regulates NSC migration is rather indirect or functionally exclusively based on somewhat artificial in vitro scratch assays.

Thus, this is an interesting study, specialized readers will benefit from the (partially) new data. Unfortunately, the conceptual advance provided appears to be somewhat limited.

February 19, 2024

Re: Life Science Alliance manuscript #LSA-2024-02659-T

Dr. Elizabeth D Kirby
The Ohio State University
1835 Neil Ave, Psychology 55
Columbus, OH 43210-1351

Dear Dr. Kirby,

Thank you for submitting your manuscript entitled "Intracrine signaling drives neural stem cell proximity to the adult hippocampus vascular niche" to Life Science Alliance. We invite further consideration of this manuscript at LSA pending the following revisions:

- Address Reviewer 1's comments. The points regarding claims made around the role of intracellular VEGFR2 and sheddases can be addressed by toning down these claims, including in Figure 10.
- Address Reviewer 2's comments. It is not necessary to show that VEGFR surface shedding is required in vivo.

Thank you for this interesting contribution to Life Science Alliance. We are looking forward to receiving your revised manuscript.

Sincerely,

- A letter addressing the reviewers' comments point by point.
- An editable version of the final text (.DOC or .DOCX) is needed for copyediting (no PDFs).
- High-resolution figure, supplementary figure and video files uploaded as individual files: See our detailed guidelines for preparing your production-ready images, <https://www.life-science-alliance.org/authors>
- Summary blurb (enter in submission system): A short text summarizing in a single sentence the study (max. 200 characters including spaces). This text is used in conjunction with the titles of papers, hence should be informative and complementary to the title and running title. It should describe the context and significance of the findings for a general readership; it should be written in the present tense and refer to the work in the third person. Author names should not be mentioned.
- By submitting a revision, you attest that you are aware of our payment policies found here: <https://www.life-science-alliance.org/copyright-license-fee>

B. MANUSCRIPT ORGANIZATION AND FORMATTING:

We thank the reviewers for their in-depth comments and appreciate the editorial staff for stewarding this transfer and revision. A common concern among reviewers was the strength of evidence for intracrine signaling. After consulting with editors, we have undertaken a major re-organization to eliminate those claims and all the data we interpreted as supporting them. In the revised manuscript, we present a much simpler story—that autocrine VEGF is necessary to maintain NSPC proximity to blood vessels, with a likely role for NSPC motility and adherence in that maintenance. Below we address individual concerns, including noting where concerns no longer apply due to removed data and claims.

Reviewer 1:

Neural stem cells (NSCs) and their progeny (IPC), (all together NSPCs), primarily locate in the subventricular zone (SVZ) and the dentate gyrus (DG) of the hippocampus. The vasculature is a key component of adult brain NSCs niches. NSCs reside in close contact with a dense capillary network in the adult mammalian hippocampus. Crosstalk between adult NSCs and the vasculature within its niche is essential for preservation of neurogenesis through adulthood. Therefore, understanding the communication between NSCs and the nearby endothelium may shed light onto how stemness is maintain and modulated within CNS neurogenic niches, which has obvious therapeutic implications for brain repair.

In this Manuscript, through a combination of in vivo experiments in mouse transgenic lines, biochemistry and in vitro work, Dause et al propose a model where intracrine VEGF-VEGFR2 signaling in NSCs is required to keep them close to blood vessels. To support this hypothesis the authors firstly used a VEGF-GFP mouse model to validate the expression of VEGF in NSPCs. Then, through ablation of VEGF in NSPCs (using a NesCreERT2; Vegflox/lox mouse), the authors found that NSPC proximity to vessels is disrupted upon NSPC-VEGF loss, apparently by NSPCs undergoing defective migration and impaired cell adhesion. The authors also indicate that VEGF receptors (VEGFR2) are not exposed in the plasma membrane of VEGF-sensitive NSPCs, but instead they exist in intracellular pools. Based in these findings the authors conclude that NSPCs regulate their proximity to the vasculature, and thus their stemness, through cell-autonomous regulation by an intracrine VEGF/VEGFR2/Akt-dependent signalling.

Although the idea is interesting, the two major findings of this work (namely disruption of NSPC proximity to vessels upon VEGF loss and intracrine VEGF/VEGFR2/Akt signalling), are not convincingly supported by the experimental data presented in the study.

Major concerns:

Recombination Efficiency in the iKD Model

> Neither in Figure 2A, E, F and H (VEGFlox/lox recombination) nor later in Figure 8B (lenti-based VEGF knockdown) did the authors prove the actual reduction of VEGF expression in NSPCs. This is specially important for interpreting the following data:

> Graphs in Figures 2C, 2D, and 2G: The data points representing the distance of RGL-NSCs and IPCs to the nearest vessels shows that the distribution of the majority of the iKD samples is similar as the WT ones. It is the case that only 4 iKD points (out of the 12 analyzed) appear to be the ones driving the statistical significance in this analysis. Therefore, if recombination

efficiency was the same in all samples, this brings forward doubts on the relevance/strength of the authors claim.

We agree that it would be ideal if we could verify loss of VEGF in individual NSPCs of each analyzed mouse. Unfortunately, VEGF immunolabeling in tissue slices reveals VEGF in the extracellular space, bound to receptor-expressing cells, internalized in receptor-expressing cells and in cells actually expressing VEGF protein themselves. Immunolabeling therefore cannot tell us about VEGF loss in individual cells. In situ hybridization could reveal this information but our tissue was not processed in a way to allow such analysis.

We have verified both models used in the present work to reduce global VEGF. For the NestinCreERT2;VEGFlox model, we verified effective knockdown following TAM using both real time PCR and ELISA of whole DG tissue in previous work (Kirby et al., 2015). To recognize the reviewer's point that we have not re-verified this knockdown in these specific mice, we have added a sentence to the results section.

“Our own work (Dause and Kirby, 2020; Kirby et al., 2015; Smith et al., 2022) and work by others (Lagace et al., 2007; Sun et al., 2014) have shown that this NestinCreER^{T2} line drives loxP recombination with high specificity in NSCs and IPCs. Though we were not able to verify VEGF knockdown in individual NSPCs here due to technical limitations, we have previously shown that TAM administration in these mice results in loss of about 1/3 of total DG VEGF compared to wildtype littermates (Kirby *et al*, 2015).”

As for the data points and their spread, we have applied an outlier test to the data in question and none were identified. We disagree with the reviewer's assertion that 4 points drive the whole analysis. If we remove the 4 points that are highest in iKD mice (and then also the 4 highest in the WT for equality), we still see a significant effect in RGL-NSC and IPC distance to vessels. Estimation plots of this hypothetical test of our data's robustness are below. With the 4 highest mice moved in both groups, both RGL-NSCs and IPCs retain statistically significant effects of genotype and even the estimated size of the difference between the means remains very similar.

RGL-NSC:
Estimation Plot with all data
p = 0.044

RGL-NSC:
Estimation Plot with 4 highest mice removed
p = 0.043

IPC
Estimation Plot with all data
p = 0.009

IPC
Estimation Plot with 4 highest mice removed
p = 0.009

As for the shRNA data, we did confirm global loss of VEGF using immunolabeling. This was in Fig EV5B,C in the former manuscript. We have moved this data to the main text now, to make it more evident. In addition, given our modified claims that this experiment only tells us about global VEGF loss, we hope that the reviewer and editors will agree that the quantification in individual NSPCs seems less critical here.

> Although the authors had their reasons to attempt the shRNA-based knockdown of VEGF using lentivirus, the achieved knockdown of VEGF is global (of note: direct proof of VEGF knockdown in figure 8 is missing) affecting different cell types. The influence of global VEGF removal on the final NSPC-vessel distance should not be ruled out or neglected.

This experiment has been reframed along the lines requested by the reviewer. It is now Figure 2 and is only used to draw conclusions about global VEGF loss. Also, the data supporting VEGF loss was in Fig EV5B,C in the former manuscript. We have moved this data to the main text now, reporting the mean and variance of VEGF immunolabeling in DG, to make these data more evident.

No experimental data to support the proposed molecular mechanism of VEGF secretion and intracellular signalling

These claims and the data related to them are deleted.

> By showing that VEGFR2 is not expressed in the plasma membrane of VEGF-sensitive NSPCs but intracellularly instead, the authors propose a model - where all molecular steps are indicated in their model figure - where NSPCs can autonomously activate an intracrine VEGF/VEGFR2/Akt signalling to modulate their distance and association with vessels. Important aspects of the diagram in Figure 10 imply mechanisms that were not experimentally confirmed in the present study.

These claims and the data related to them are deleted.

> Crucial to this intracrine signalling, is the removal of plasma membrane-exposed VEGFR2 by shedases. However, the only piece of the proposed mechanism that is experimentally addressed in this work is the involvement of Akt downstream of VEGFR2 activation. The authors do not explore any aspect of the subcellular localization or sorting of either VEGF or VEGFR2, or how do they come together intracellularly for the activation of the signal. Similar situation comes to the regulation of the shedases that ultimately allow to favor intracrine vs. autocrine/paracrine signalling. Moreover, no molecular mechanism is proposed for the apparent deregulation of migration and cell adhesion proposed to be the driver of the observed phenotype.

These claims and the data related to them are deleted. We leave much of this as description of future work needed.

Points requiring further clarification or improvement

> The technical approaches to support some of the arguments are not clear.

We are uncertain whether this comment has been addressed by the reorganization of the manuscript or not. We remain open to further guidance on this issue.

> In Figure 1F, RGL-NSCs are differentiated from astrocytes based on their morphologies to measure the Vegfa expression in both cell types. An inclusion of astrocyte markers such as Sox9 could have better differentiated both cell types.

We have added this point to the limitations section. We agree with the reviewer on this point and it is why we refer to these cells as putative RGLs and astrocytes in the main text. We had great technical difficulty getting some antibodies to work in conjunction with the processing necessary for RNAscope, which limited the markers we could use. This limitation is also why we

supplement this data with a reporter mouse and analysis of RNAseq datasets from the literature. We hope that these multiple sources of evidence, coupled with the new caveat noted in the limitations section are sufficiently clear about the strengths and weaknesses of our data.

“In addition, RNAscope analysis would be more certain if additional markers to differentiate astrocytes from NSCs were used.”

> The distance between CD31+ vasculature and RGL-NSCs was measured while considering the cell body center of EYFP+/GFAP+RGL-NSCs (Figure 2C). The changes in cell shape could bias this distance. A more reliable approach could be to measure the distance of the apical processes of NSCs from blood vessels.

We debated intensely how to measure distance to a vessel. Changes in cell shape figured prominently in our concerns. We resolved to use cell center because it would be less subject to variance with cell membrane shape changes. It occurs to us that we did not appropriately define this measure in the methods and this may have caused some confusion, however. Cell center identification was aided by Hoechst label, to show the nucleus. We have modified the methods to reflect this.

We considered measuring distance from apical processes to vessels but we did not feel confident in the replicability of this measure—where exactly a RGL process ended could be difficult to define sometimes and was exceedingly difficult to standardize such that multiple observers would be able to generate similar data. Instead, we chose the measure of a yes/no decision of whether we saw RGL apical processes or terminals intersect with blood vessels (old Fig 2I, new Fig 3I). In that sense, this measure was somewhat based on the idea of Scholl analysis, where we decided whether we saw an intersection of EYFP+/GFAP+ processes with a CD31+ blood vessel.

We have added more discussion of this limitation to the limitations section:

“In the present work, we used distance from cell center, plus assessment of whether RGL-NSC processes appeared to intersect with a blood vessel as measures of vascular association. Neither of these measures takes into appreciation the full 3D shape of the cells and how they contact vessels. 3D imaging of thick, cleared tissue samples may also provide a more nuanced view of vascular structure and NSPC relationship to that vasculature.”

> The authors have shown CD31 positive cells as individual dots instead of an intact vascular structure (Figure 2J). Similarly, in Figure 3E, CD31 staining is inconsistent in both wild type and iKD. Thus, conclusions drawn just from these figures are questionable.

There is a misunderstanding what we were pointing to in Fig 2J. The dots are background (common in this stain). We have modified the image to highlight 3 points of potential contact between RGL-NSC process and a long vessel running at an angle along the top of the image. We have also added a sentence to the methods to better note that we selected for elongated structures, not punctate background. We have adjusted to brightness and contrast in Fig 3E to make the vessels more readily visible.

“Vessels were defined as elongated structures, distinct from punctate background.”

> In Figure 4B, the cellular population from the wild type shows almost no astrocytes. If this is true, how is astrocyte-associated gene analysis from wild type compared with iKD?

We did not intend to draw any conclusions from comparisons between Wt and iKD outside of the NSPC clusters described in the main text. We agree that comparing Wt and iKD within the astrocyte cluster would be inappropriate given how few astrocytes were identified. We have reviewed Table S1 and the main text but we are not seeing where the reviewer finds this comparison within the astrocyte cluster. If we have said anything that implies that we made such a comparison, we would appreciate some guidance so we can delete it.

> Fig 4E: The information intended to be displayed in the heat map should be further explained. It has all the cell types separately for WT and iKD, but the heatmap scale says "log₂(Foldchange)" - fold change of what over what?

This has been clarified in the figure legend: "Values are log₂ fold change of the average normalized transcript count in a group over the cluster average."

> Figure 8C shows no statistical significance between scramble and Vegfa lentiviruses vectors. Does it show the inefficiency of Vegfa shRNA in removing Vegfa significantly from GFP+ NSCs? Moreover, upon Vegfa treatment, GFAP projection is also altered upon VEGF depletion in RGL-NSCs (Fig 8E).

These data and images are deleted from the revised manuscript.

Referee #2:

In this manuscript Dause et al. address the role of VEGF in the maintenance and positioning of adult neural stem cells relative to the vasculature. This is an extension of previous work by the senior author showing that VEGF plays a role in the maintenance of hippocampal neuronal stem cells. The authors used a combination of conditional gene ablation of VEGF and in vitro experiments and claim that VEGF plays a role in maintaining neural stem cells in close proximity to the vasculature but works through a cell-autonomous mechanisms rather than through paracrine signalling. Although the question of whether VEGF plays a role on hippocampal neural stem cell maintenance and differentiation is not novel, the authors propose that loss of VEGF affects an intracellular autocrine signal that results in putative neural stem cells moving away from the peri-niche vasculature in the subgranular layer of the dentate gyrus. The authors generated considerable data quantifying changes in location of GFAP+ putative radial stem cells in the dentate gyrus relative to vascular cells. They then go on to perform single cell sequencing of cells isolated from the dentate gyrus of Vegfa wildtype and Vegfa conditional knockout mice. They identify changes in gene expression associated with cell adhesion and cell attachment following Vegfa ablation. They then test the role of VEGF in neural progenitor migration in scratch assays in vitro using siRNA knockdown approaches. They show that VEGF seems to work through VEGFR2 in an intracellular signalling pathway and that VEGFR2 is not expressed at the cell surface on hippocampal stem cells and seems to be released from the surface by a metalloprotease. The analysis is interesting but mainly descriptive and does not mechanistically go beyond the previous indication that VEGF could regulate neural stem cell maintenance. There are no really insights into the intracellular signalling pathway activated by VEGF and how

this regulates neural stem cell behaviour. It is unclear which metalloprotease potentially regulates VEGFR shedding and whether this has a biological function in vivo.

Major concerns

The in vivo "location" defects detected in the VEGF conditional knockout mouse are rather modest. The cells move 2-3 μ m further away from the blood vessels (~14.5 μ m for control neural stem cells and 17 μ m for Vegfa conditional deleted cells). It is not clear and not addressed whether this movement is important for the changes in proliferation and the maintenance defects caused by Vegfa deletion.

The changes in distance are not large, we agree. Indeed, we don't see how they could be large given the size of the SGZ, unless the NSPCs left the SGZ entirely. We express the changes relative to distance of a random cell to emphasize a way to consider the magnitude of movement of cells in such a narrow space. The preferential association of NSPCs with vessels (i.e. greater than random) has long been hypothesized to be important. In the case of IPCs, we show that this preferential association is completely lost after loss of NSPC-VEGF. Such a complete loss apparently only requires a movement of ~2 μ m in this tissue. We agree that we cannot prove whether this loss is meaningful, just as no previous work has thus far proved that the existence of preferential association is meaningful to begin with. Our work might have provided a window into this open question, as we have reported loss of NSC maintenance in VEGF iKD mice. But this conclusion is thwarted by the existence of self-signaling via VEGFR2 to maintain stemness even in vitro. In vivo, we cannot tell the difference between effects due to loss of vascular proximity and effects due to loss of self-VEGFR signaling that directly supports stemness. We have added more about this limitation in the new discussion.

"The mature DG vascular niche is widely hypothesized to support adult neurogenesis, in particular via exposure to circulatory factors (Karakatsani *et al*, 2019). Beneficial blood-borne factors, such as those associated with exercise (Fabel *et al*, 2003a; Horowitz *et al*, 2020; Trejo *et al*, 2001), as well as detrimental factors, such as those associated with old age (Villeda *et al*, 2011), drive changes in NSPC proliferation that result in altered neurogenesis and correlated improvements or impairments, respectively, in hippocampus-dependent memory function. The unique association of NSPCs with local capillaries is frequently cited as a likely mechanism for how circulating factors drive robust changes in NSPC proliferation and thereby neurogenesis (Kim *et al*, 2021a; Licht *et al*, 2020b). Our findings therefore highlight a new potential mechanism by which NSPC-derived VEGF could support maintenance of NSCs and therefore adult neurogenesis. Though the distance-to-vessel changes observed following loss of NSPC-VEGF in the present study were often small in magnitude (2-3 μ m), in IPCs, this difference was enough to cause a complete loss of preferential association beyond that found for a random SGZ cell. The functional relevance of loss of NSPC preferential association with vessels remains unclear, though. While we previously showed that NSPC-VEGF loss impairs NSC maintenance (Kirby *et al*, 2015), it is important to note that with this model, we cannot discern whether that effect is due to loss of vascular proximity or loss of direct VEGF receptor signaling to support stemness independent of vascular signals. We also cannot be sure whether any of the changes in relationship to vasculature here are a cause of loss of stemness or a consequence of it."

In Figure 2I the author show loss of putative stem cell associate with blood vessels. However, it is not clear what percent RGL processes contacting vasculature really means and whether this

is relevant. It is also unclear if this phenotype is a cause of neural stem cell defects or as a result of changes in the neural stem cell maintenance as a result of Vegfa deletion.

The percent of RGL processes contacting vasculature was intended as a measure of NSC association with vasculature. For example, in Tang et al. (2015), cell body contact with vessels is used as a primary measure of the vascular niche. We similarly use distance from cell body to vessel for much of our analysis, but this measure fails to capture the unique aspect of the NSC apical process having terminals that contact blood vessels. Thus, we included RGL process contact. We have added further discussion of these uncertainties raised by the reviewer in the discussion (see paragraph above).

Then authors show that loss of VEGF expression by Nestin::CreERT2 expressing cells and their progeny does not affect the vascular structure in the dentate gyrus but the graphs in Figure 3D and 3 F have statistics which is confusing. This just reflects vascular density differences across the different regions of the dentate gyrus. These data do not bring much to the main manuscript.

We can see how the way the stats were displayed in this figure was confusing. Both genotypes are shown but none of the comparisons between genotypes were significantly different. There were differences between layers of DG, which we tried to indicate but it did admittedly get confusing. We have removed those markers of significance and leave them for the supplementary table of statistical tests instead. We do feel the data are essential, though, as a primary question when considering the loss of NSPC-vessel association in VEGF iKD mice would be whether there were gross changes in the vasculature—death, proliferation, vessel coverage/migration. The data in this figure address this question.

The single cell data analysis shows relatively minor effects on gene expression. In Figure 4C it is not clear what is meant by Average Expression - is this a Z-score?

Fig 4C: This is a standard measure output by Seurat. It is a z-score. This detail has been added to the figure legend.

In addition, the fold changes shown in the heat map in Figure 4E are very low (Log2 fold change of +/- 0.1).

First, we think we did not sufficiently explain the scale used. The Log2Fold change is the change of a group versus the whole cluster average. So, a gene at -0.2 log2fold change for iKD is -0.2 log2fold change from the whole cluster average; the paired Wt sample is therefore around +0.2 log2foldchange from the cluster average. If we display the fold change data as fold change over wild type, the whole wt side of the heat map is one solid color (=0). We can display the data that way, if desired. It looks a little odd in our opinion though.

Aside from the confusion about the scale, we still agree that the size of the change in this data are not huge. However, this is a common feature of single cell RNASeq data, when it is used to identify differentially expressed genes (DEGs) between prospectively defined groups. The nature of the input data and analysis in single cell RNAseq tends to yield genes that are very highly expressed and have low fold changes. As a useful demonstration, we performed a study recently where we subjected cells from the exact same samples to both single cell RNAseq and bulk RNAseq (Denninger et al 2022). From the same cells (just split in half after collection), the DEGs identified were quite different between the two workflows and the single cell data yielded genes that were much more highly expressed and showed lower fold changes. Similar observations are also documented by others (Squair et al 2021). Importantly, we found in our previous work that DEGs from single cell and bulk analysis were both replicable, and that they

yielded highly overlapping GO processes, suggesting that the bias in types of DEGs could be overcome when looking more broadly at what processes those genes might point to.

Again the description of the analysis and quantification are not clear and I assume this is a Z-score of the fold change across all cell types.

The measure used here has been further clarified in the figure legend: "Values are \log_2 fold change of the average normalized transcript count in a group over the cluster average."

How many samples were analyzed here and how many cells are contribute to the gene expression heat map in each category?

For the number of mice, this was in the methods (5/genotype), but it should have also been in the figure legend. We have fixed this omission and added the mouse numbers to the Fig 5 legend. The information about cell numbers/cluster/genotype is available in supplementary table 1 in a tab titled "Raw Cell Counts". We are not sure whether the reviewers received this table or not. We would appreciate guidance from the journal if there is any trouble sharing it. It is an excel table and the amount of information in it does not translate well into a Word doc format.

Figure 5: The scratch assay is interesting but the relevance of this for neural stem cells that are sedentary in vivo is not clear.

We agree that a model of quiescent NSCs would be useful. We have added that data, using the well-established BMP4 model of reversible quiescence. We have also added a cell attachment assay in both proliferative and quiescent NSCs to further explore the role of cell adhesion. These data are added to Figure 6. In short, we found that quiescent NSCs are largely stationary (consistent with previous work and in vivo), but that they show reliance on VEGFR2 signaling for cell adhesion. Proliferative NSCs also showed reliance on VEGFR2 signaling for cell adhesion.

The labels of the axes of the graphs (in addition in most of the figures) is not clear and the authors need to explain what is exactly measured and the units of the y-axis!. Why does the control virus affect the migration of infected cells?

The axes have been further explained in the methods. We also modified the scratch assay axis slightly, putting it on a 0-1 scale where 0 equates to no ingression and 1 equates to complete ingression. Nothing about the statistical comparisons has changed—this is only a change in how the data are displayed to make them easier to understand. In short, each scratch was unique in size because it was made manually. Therefore, the measures used are normalized to initial scratch size and then inverted to make ingression be reflected as increasing numbers. Because the SU5416 assay was a whole well treatment, this was done using total scratch area.

Here is the new verbatim text from the methods. For the SU5416 treatment, "Because each scratch was unique, areas for each scratch were individually normalized to the area at 0h. Scratch ingression was calculated as 1- (scratch area at hour X/scratch area at 0h). On this scale, 0 represents no ingression and 1 represents complete scratch closure."

The shRNA data has been removed from this figure because without the intracrine half of our story, it does not make sense to measure movement of just GFP+ cells here, while whole scratch area was measured in the SU5416 assay.

The images in Figure 5B and 5 D are not visible.

We think this might be due to the lower resolutions used in these peer review formats. We hope

that using the higher resolution images now being uploaded with revision will resolve this problem. Fig 6C is the new number for the relevant image that has been retained.

The authors should show if the surface sheading of VEGFR is required in vivo.

No longer applicable as the claims related to signaling mechanism have been eliminated.

In Figure 7F, the authors need to explain why TAPI treatment affects AKT activity in the absence of VEGF if the assay should report the intracellular VEGF activity? Does TAPI treatment increase the AKT levels in response to VEGF treatment higher than the VEGF treatment alone?

No longer applicable as the data and claims related to signaling mechanism have been eliminated.

Why are the distances of stem cells from the nearest vessel different in the controls in Figure 7C compared to the values in Figure 2C? This is particularly true for the Vegfa siRNA GFP- cells which should be wildtype or not?

The absolute value of the distances will depend on tissue processing and storage history, which can vary from cohort to cohort. The distances we are measuring are small, as the reviewer notes, and there are numerous variables that could create differences when comparing across cohorts that were housed, harvested, fixed and processed separately. These two experiments, for example, were performed years apart from each other. One was in a transgenic line that had been bred in-house for multiple generations, and the other was in Wt mice directly from Jackson labs. All tissue processing was completely separate between experiments and used separate batches of all solutions during tissue harvest, cryopreservation and immunolabeling. We always compare experimental mice to controls processed in parallel to control for any potential variances across cohorts.

Referee #3:

In their manuscript «Intracrine signaling drives neural stem cell proximity to the adult hippocampus vascular niche» Dause and colleagues aimed to characterize the mechanisms how neural stem cell (NSC)-derived VEGF affects neurogenesis in the adult mouse hippocampus. The work presented is a continuation of previous work from Kirby et al., 2015 (PNAS) that had described the expression of VEGF in hippocampal NSCs and the functional consequences of NSC-specific VEGF genetic deletion.

The experimental design of the current study is straightforward, the data are convincing.

However, the advance provided is unfortunately somewhat limited.

There are clearly more in-depth analyses compared to the Kirby et al 2015 study but those new data are largely "details". Indeed, the 2015 PNAS used similar approaches (e.g., VEGF-GFP mice) to analyze expression of VEGF (Kirby Figure 1 is somewhat similar to the current Figure 1), genetic deletion of VEGF was done using the "same" NestinCreERT2 mice in the current and previous study, vascular density was analyzed in the 2015 study using CD31 densities (as it was done in the current study). There is no question that the current ms contains novel data (e.g., distances from vessels of NSCs; even though this may be also influenced by the increase of proliferative cells upon VEGF-deletion) and subsequent molecular analyses, e.g., using transcriptomics. However, the interpretation that the intracrine loop (it is not entirely clear why the authors suggest that the VEGF loop is intracrine rather than autocrine; given their 2015 data suggesting that NSCs secrete VEGF in substantial amounts) regulates NSC migration is rather

indirect or functionally exclusively based on somewhat artificial in vitro scratch assays. Thus, this is an interesting study, specialized readers will benefit from the (partially) new data. Unfortunately, the conceptual advance provided appears to be somewhat limited.

We hope the change of journal venue satisfies the reviewer in terms of being appropriately placed for the interested audience.

April 4, 2024

RE: Life Science Alliance Manuscript #LSA-2024-02659-TR

Dr. Elizabeth D Kirby
The Ohio State University
1835 Neil Ave, Psychology 55
Columbus, OH 43210-1351

Dear Dr. Kirby,

Thank you for submitting your revised manuscript entitled "Autocrine VEGF drives neural stem cell proximity to the adult hippocampus vascular niche". We would be happy to publish your paper in Life Science Alliance pending final revisions necessary to meet our formatting guidelines.

- please be sure that the authorship listing and order is correct
- please list first the main figure captions and then the supplementary figure captions after them
- please add a callout for Figure 4F and 6F-G to your main manuscript text

A. FINAL FILES:

B. MANUSCRIPT ORGANIZATION AND FORMATTING:

****It is Life Science Alliance policy that if requested, original data images must be made available to the editors. Failure to provide original images upon request will result in unavoidable delays in publication. Please ensure that you have access to all original**

data images prior to final submission.**

The license to publish form must be signed before your manuscript can be sent to production. A link to the electronic license to publish form will be available to the corresponding author only. Please take a moment to check your funder requirements.

Sincerely,

April 8, 2024

RE: Life Science Alliance Manuscript #LSA-2024-02659-TRR

Dr. Elizabeth D Kirby
The Ohio State University
1835 Neil Ave, Psychology 55
Columbus, OH 43210-1351

Dear Dr. Kirby,

Thank you for submitting your Research Article entitled "Autocrine VEGF drives neural stem cell proximity to the adult hippocampus vascular niche". It is a pleasure to let you know that your manuscript is now accepted for publication in Life Science Alliance. Congratulations on this interesting work.

DISTRIBUTION OF MATERIALS:

Again, congratulations on a very nice paper. I hope you found the review process to be constructive and are pleased with how the manuscript was handled editorially. We look forward to future exciting submissions from your lab.

Sincerely,
